

# Tracking down the lizards from Gravenhorst's collection at the University of Wrocław: type specimens of *Callopistes maculatus* Gravenhorst, 1838 and three *Liolaemus* species rediscovered

Bartosz Borczyk* and Tomasz Skawiński*

Department of Evolutionary Biology and Conservation of Vertebrates, University of Wrocław, Wrocław, Poland
* These authors contributed equally to this work.

## ABSTRACT

Johann Ludwig Christian Gravenhorst's herpetological collection at the Museum of Natural History, University of Wrocław included numerous important specimens of amphibians and reptiles. The majority, if not the entirety, of this collection has long been thought to be lost. However, we were able to rediscover some type specimens of lizards. The rediscovered specimens include the holotypes of *Liolaemus conspersus* and *L. hieroglyphicus*, one syntype of *Callopistes maculatus* (here designated as the lectotype) and two syntypes of *L. lineatus* (one of which is herein designated as the lectotype). Reexamination of these specimens indicates that previous synonymies proposed for *L. conspersus* and two syntypes of *L. hieroglyphicus* are problematic; furthermore, more complex taxonomic work is needed to resolve this issue. Two rediscovered syntypes of *L. lineatus* differ in several scalation traits and are possibly not conspecific. The type specimens of several other species of lizards from Gravenhorst's collection (*Liolaemus marmoratus*, *L. unicolor* and two other syntypes of *L. lineatus*, *Leiocephalus schreibersii* and *Chalcides viridanus*) were not found and are probably lost.

Corresponding authors
Bartosz Borczyk,
bartosz.borczyk@uwr.edu.pl
Tomasz Skawiński,
tomasz.skawinski@uwr.edu.pl

## INTRODUCTION

German naturalist Johann Ludwig Christian Gravenhorst (1777–1857) was the founder and first director of the Zoological Museum at the University of Wrocław (Zoologisches Museum der Universität Breslau, currently the Museum of Natural History). His main interests lay in entomology, particularly beetles and, later, ichneumonid wasps (*Jałoszyński & Wanat, 2014*). However, like many naturalists of his time, Gravenhorst also had a comprehensive knowledge of many other groups of animals. During his directorship at the museum (from its founding in 1814 to Gravenhorst's death in 1857), he acquired large collections of fishes, amphibians and reptiles from around the world, and published numerous articles on the latter two groups of vertebrates (*Jałoszyński & Wanat, 2014*);

among them were descriptions of several new species of reptiles (*Gravenhorst, 1838*; *Gravenhorst, 1851*). The type specimens of these forms were usually deposited in the museum in Wrocław; however, at least half of the zoological specimens and 90% of exhibits were destroyed during the World War II, particularly when the city was sieged and turned into Festung Breslau in 1945 (*Wanat & Pokryszko, 2014*). After the war, the identification of some presumably lost specimens was hindered by the fact that—as a part of the 'polonisation' of the museum, when the city became part of Poland again—original German labels were replaced by Polish labels (*Wiktor, 1997*); in the process, errors (e.g. misspellings) were sometimes made (*Borczyk, 2013*). Some of the important Gravenhorst's zoological specimens were rediscovered after the war (*Jałoszyński & Wanat, 2014*), but it was long thought that most or even all of the type specimens of reptiles were also lost at that time (*Dubois & Ohler, 2000*; *Nori, Abdala & Scrocchi, 2010*; *Etheridge & Frost, 2012*). However, it turned out that some of them survived the war. First, the holotype of *Liolaemus lemniscatus Gravenhorst, 1838*, was recently redescribed (*Borczyk, 2013*).

Many species of *Liolaemus*, including all those named by Gravenhorst (except *L. lemniscatus*) were synonymised by *Boulenger (1885)*. He did not discuss his nomenclatural decisions but they were later accepted, even if only tentatively, by *Etheridge & Espinoza (2000)*, *Etheridge & Frost (2012)*, *Pincheira-Donoso, Scolaro & Sura (2008)* and *Abdala & Quinteros (2014)*. However, some other species considered junior synonyms by *Boulenger (1885)*, such as *L. oxycephalus* (*Wiegmann, 1834*) and *L. inconspicuus* (*Gray, 1845*), were later revived from synonymy (*Troncoso-Palacios & Garin, 2013*), which warrants a careful re-examination and comparison of these historical specimens. Here, we redescribe type specimens of lizards that were rediscovered in the collections of the Museum of Natural History, University of Wrocław. We also discuss taxonomic implications of these findings and provide a catalogue of type specimens of all lizard species named by Gravenhorst.

## MATERIALS AND METHODS

We focused on describing the morphological characters that were used by previous authors (*Ortiz, 1981*; *Laurent, 1992*; *Lobo, 2001*, *2005*; *Troncoso-Palacios & Garin, 2013*; *Troncoso-Palacios et al., 2015a*) and considered to be taxonomically informative. The identification and nomenclature of scales follow *Etheridge (2000)* for liolaemids and *Harvey, Ugueto & Gutberlet (2012)* for teiid *Callopistes maculatus*. The colour pattern description follows the terminology given by *Lobo & Espinoza (1999)*. Whenever possible, measurements were taken using a digital caliper to the nearest 0.1 mm. When specimens were strongly curled and it was impossible to make a linear measurement, the snout-vent length (SVL) was measured using a string reaching along the midventral surface from the snout to the vent.

A small amount of muscle tissue was taken from putative holotypes of *Liolaemus conspersus* and *L. hieroglyphicus*, and two specimens of *Chalcides viridanus* collected by Zimmer (other rediscovered specimens were in too poor condition) in an attempt to extract DNA. Unfortunately, these attempts were unsuccessful, so all descriptions and taxonomic assessments were made entirely based on morphology.

The electronic version of this article in portable document format will represent a published work according to the *International Commission on Zoological Nomenclature (1999)* (ICZN), and hence the nomenclatural acts contained in the electronic version are effectively published under that Code from the electronic edition alone. This published work has been registered in ZooBank, the online registration system for the ICZN. The ZooBank life science identifiers (LSIDs) can be resolved and the associated information viewed through any standard web browser by appending the LSID to the prefix http://zoobank.org/. The LSID for this publication is: urn:lsid:zoobank.org: pub:3CDA8E5A-30C8-4847-B4A4-C644595361FC. The online version of this work is archived and available from the following digital repositories: PeerJ, PubMed Central and CLOCKSS.

## RESULTS

### Extant types

Reptilia *Laurenti, 1768*
Squamata *Oppel, 1811*
Teiidae *Gray, 1827*

*Callopistes maculatus Gravenhorst, 1838*

**Original name.** *Callopistes maculatus Gravenhorst, 1838*: 744.

**Type specimens.** Two syntypes (described by *Gravenhorst (1838)* as 'larger' and 'smaller' ones) collected by F.S. Scholtz 'at the foot of the Cordilleras' (i.e. in the Andean foothills). One of these specimens (MNHUW 1320, male, SVL 141 mm)—here identified as the larger one, based on the measurements given by *Gravenhorst (1838)*—still exists in Wrocław.

**Present name.** *Callopistes maculatus Gravenhorst, 1838*.

**Remarks.** *Gravenhorst (1838)* described the teiid *Callopistes maculatus* on the basis of two syntypes collected by F.S. Scholtz. *Callopistes maculatus* is the type species of the genus *Callopistes Gravenhorst, 1838*, which itself is a type genus of the subfamily Callopistinae *Harvey, Ugueto & Gutberlet, 2012*. However, the phylogenetic position of *Callopistes* is not well understood; *Tucker et al. (2016)* placed it within Tupinambinae (though as sister to all other tupinambines), thus not recognising Callopistinae, and so did *Brizuela & Albino (2017)*, though some other authors retained that name (*Goicoechea et al., 2016*; *Quadros, Chafrat & Zaher, 2018*).

The lizard currently known as *Callopistes maculatus* was first described by *Molina (1782)* under the name *Lacerta palluma*. *Gravenhorst (1838)* misidentified the species, confusing it with a strikingly different iguanian lizard, for which he coined the new generic name *Phymaturus*. Despite this, *Gravenhorst's (1838)* application of these two names became universal. *Veloso, Núñez & Cei (2000)* proposed the combinations *Callopistes palluma* (for current *Callopistes maculatus*) and *Phymaturus flagellifer*

(formerly *Centrura flagellifer*, currently *Phymaturus palluma*). They attempted to synonymise *Callopistes maculatus* with *Callopistes palluma* and designated a neotype (MNHNCL 2909) for the latter name. However, these proposals were later rejected (*Etheridge & Savage, 2003*; *International Commission on Zoological Nomenclature, 2005*). *Callopistes maculatus* is a valid name for this teiid lizard and its type specimens are those described by *Gravenhorst (1838)*.

One rediscovered teiid lizard specimen exactly matches the pattern of cephalic scales illustrated by *Gravenhorst (1838)*, so we regard it as the rediscovered syntype. The second specimen from the type series was not illustrated and has not been found. However, according to *Gravenhorst's (1838)* description, it differed from MNHUW 1320 in several scale counts. Moreover, the type locality of *Callopistes maculatus* is so vague that it cannot be unambiguously stated that both syntypes were from the same locality. Because two other subspecies of *Callopistes maculatus* occur in Chile (*Donoso-Barros, 1960*), for the sake of stability, we designate the surviving specimen as the lectotype.

The lectotype is a dry specimen with a broken tail, with the pieces stored together. The right forelimb is also broken, but remains attached to the body (Figs. 1 and 2). Some aspects of the animal morphology are distorted because of soft tissue shrinkage. However, it seems that the postcloacal buttons are present, which indicates that the lectotype is a male. The dentition is heterodont, with anterior teeth monocuspid, conical or slightly recurved, and posterior teeth with two, sometimes three, cusps. At least two palatal teeth are present on the right pterygoid and at least one tooth is present on the left one. The tongue is deeply bifurcated. The interparietal scale is small, roughly hexagonal and surrounded by five scales, including only two parietals. A total of 12 scales separate the interparietal from the rostral and eight scales separate the rostral from the frontal (at the midline). The rostral scale is separated from the nasal by one scale and is twice as wide as it is long. There are eight supralabials. The frontal scale is roughly pentagonal, with a strongly asymmetrical posterior border; it is much more concave on its right side. Two rows of long, low lorilabials are present; dorsally to these, there are three loreals. The upper temporal scales are small and oval, while the lower temporal scales are larger, with some of them hexagonal rather than oval. The nasal scale is separated from the canthal, in contact with seven scales. There are eight enlarged supraoculars on the right side and eight on the left side.

The mental scale is wider than it is long. There are 10 infralabials. Five scales contact the second infralabial (including two other infralabials). Posterior to the mental scale, there are four pairs of large chinshields. They are separated from the infralabials by one (anteriorly) or two (posteriorly) rows of sublabials. The first pair of chinshields contact at the midline, but the second and farther pairs are separated by small, oval gular scales (it is impossible to tell the exact number). The whole throat is covered by small, oval or roughly pentagonal, juxtaposed gulars, all about the same size. The interangular sulcus is absent and the intertympanic sulcus is present. Between the throat and the posterior surface of the arms, the scales are also juxtaposed, but larger; some of them are oval, elongated, pentagonal or hexagonal. The whole venter (behind the posterior end of the arms) is covered by large, rectangular, juxtaposed scales. Similar scales, though

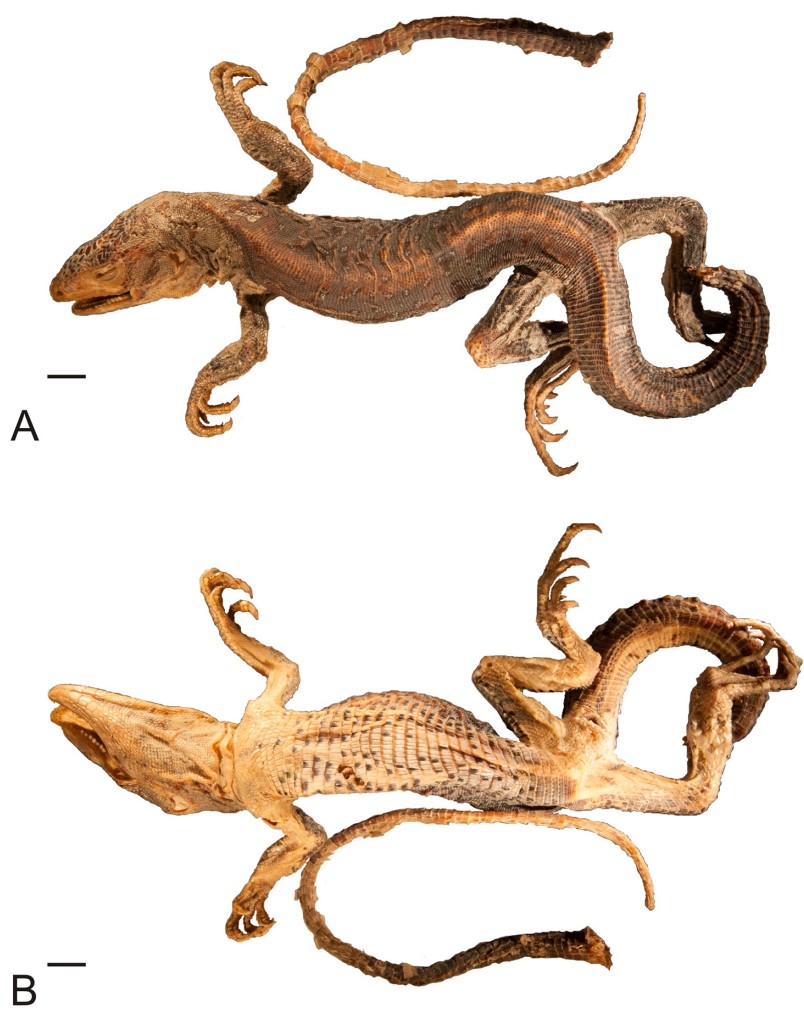

**Figure 1 Lectotype of *Callopistes maculatus Gravenhorst, 1838* (MNHUW 1320).** (A) Dorsal view. (B) Ventral view. Scale bar = 1 cm. Photographs by Bartosz Borczyk.

slightly smaller and imbricate, also cover the entire tail, forming numerous annuli. The dorsal surface of the forelimbs is covered by roughly round, juxtaposed or slightly imbricate scales. The ventral forelimb scales are much smaller, round and juxtaposed. The dorsal hindlimb scales are relatively large, quadrangular, juxtaposed or slightly imbricate. The posterior surface of the thighs is covered by much smaller, round or quadrangular, juxtaposed scales. The ventral thigh scales are roughly rectangular, juxtaposed or minimally imbricate. The ventral shank scales are much larger, rounded and juxtaposed (distally slightly imbricate). The subdigital lamellae on both the fore- and hindlimbs are impossible to count precisely because of the distortion of the specimen (see Table 1 for morphometric data on this and other redescribed lizards).

Most of the original colour pattern is not preserved. The dorsum and flanks are dark brown-reddish and the ventral body part is almost uniformly yellowish, with many dark spots on the venter. The pileus is greyish with numerous large, irregular dark spots.

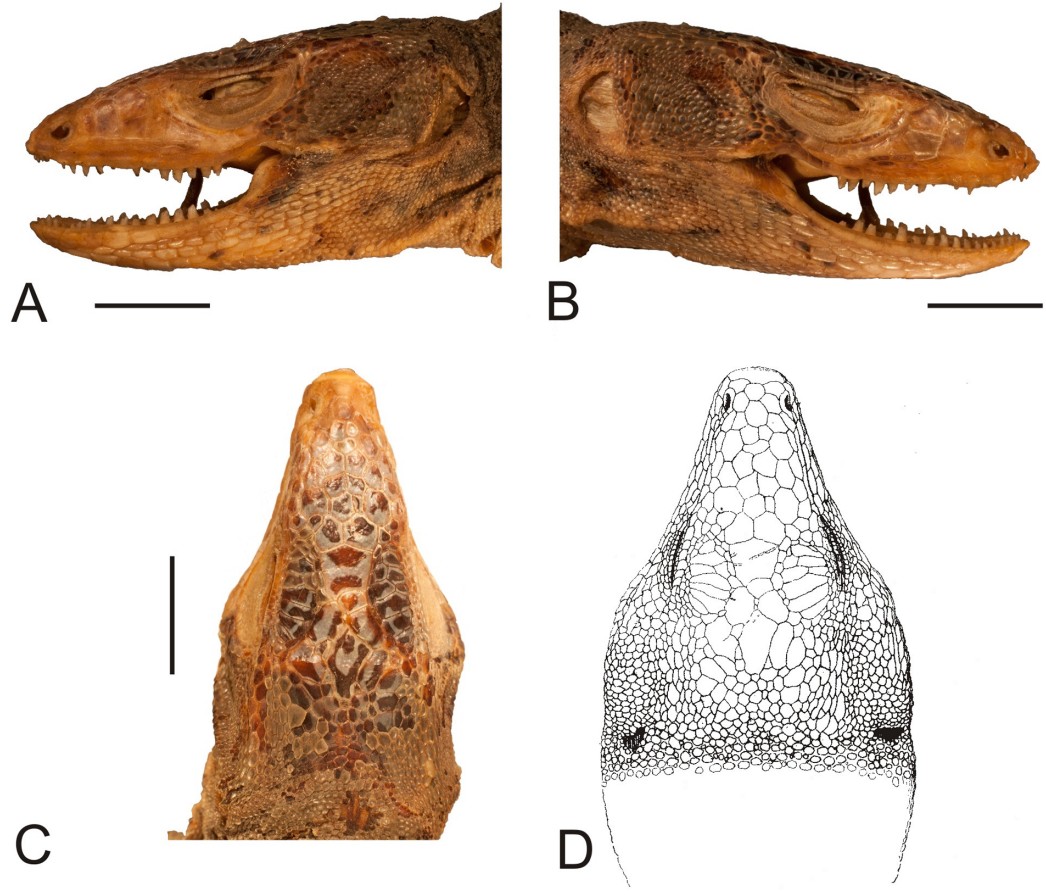

**Figure 2** **Head of the lectotype of *Callopistes maculatus* *Gravenhorst, 1838* (MNHUW 1320).** (A) Left lateral view. (B) Right lateral view. (C) Dorsal view. (D) Dorsal view as illustrated by *Gravenhorst (1838)*. Scale bar = 1 cm. Photographs by Bartosz Borczyk.

Reptilia *Laurenti, 1768*

Squamata *Oppel, 1811*

Liolaemidae *Frost & Etheridge, 1989*

*Liolaemus conspersus Gravenhorst, 1838*

**Original name.** *Liolaemus conspersus Gravenhorst, 1838*: 737.

**Type specimen.** A single specimen (MNHUW 1321, probably male, SVL 83.9 mm) collected by F.S. Scholtz in 'Cauquenes' (in the current Maule region, Chile). This specimen still exists in Wrocław.

**Present name.** *Liolaemus conspersus Gravenhorst, 1838*.

**Remarks.** This species was described by *Gravenhorst (1838)* on the basis of a single individual from Cauquenes in Chile. It was synonymised with *Ptychodeira Fitzingerii* (*Duméril & Bibron, 1837*) (currently *Liolaemus fitzingerii*) by *Fitzinger (1843)* and with *L. nigromaculatus* (*Wiegmann, 1834*) by *Boulenger (1885)*. However, *Hellmich (1934)*

**Table 1 Morphometric measurements of the rediscovered lizards.**

|  | Callopistes maculatus | Liolaemus conspersus | Liolaemus hieroglyphicus | Liolaemus lineatus second variety | Liolaemus lineatus third variety |
|---|---|---|---|---|---|
| SVL | 141 | 83.9 | 54.0 | 89.4 | 54.3 |
| Tail length | 292 (the last 203 mm are broken) | 137.7 | 11.7 (original part of the tail) | 123.4 | 76.0 |
| Head length | 38.2 | 21.0 | 13.2 | 22.8 | 18.6 |
| Head width | 20.9 | 16.9 | 9.5 | 16.1 | 13.2 |
| Head height | 19.3 | 9.0 | 6.8 | 12.9 | 11.0 |
| Axilla-groin distance | 67.0 | 40.5 | 22.6 | – | – |
| Tail base width | 14.8 | 11.8 | 7.0 | 11.6 | 9.0 |
| Interorbital distance (between postorbital semicircles) | 4.9 | 1.4 | 1.5 | 3.0 | 2.6 |
| Eye-auditory meatus distance | 12.6 | 7.2 | 4.1 | 9.1 | 6.8 |
| Internarial distance | 4.7 | 3.9 | 2.7 | 3.6 | 3.1 |
| Arm length | 19.1 L | 11.2 R/11.8 L | 7.4 R/6.2 L | 12.4 R/13.1 L | 11.1 R/10.9 L |
| Thigh length | 33.0 R/26.6 L | 14.2 R/16.0 L | 10.2 R/10.1 L | 15.6 R/16.4 L | 14.7 R/14.1 L |
| Shank length | 31.1 R/28.2 L | 17.9 R/17.7 L | 10.2 R/10.1 L | 16.8 R/17.6 L | 14.4 R/14.7 L |
| Foot length | – | 26.2 R/25.4 L | 16.1 R/14.6 L | 26.1 R/24.8 L | 19.9 R/19.0 L |
| Subocular length | – | 5.5 R | 4.2 R | 6.1 R | 5.7 R |
| Preocular length | – | 1.6 R | 0.9 R | 1.4 R | 1.3 R |
| Rostral length/width | 2.3/4.7 | 1.6/3.9 | 0.8/2.6 | 1.4/3.8 | 0.9/3.7 |
| Mental length/width | 2.6/4.1 | 2.3/4.7 | 1.5/2.7 | 1.9/3.9 | 1.7/3.3 |
| Auditory meatus height/width | 6.0/3.5 R | 4.1/2.5 R | 2.0/1.5 R | 3.5/2.8 R | – |

Note:
'R' indicates a measurement taken from the right side of the specimen and 'L' indicates a measurement taken from the left side. Note that not all measurements could be made.

was unsure about the synonymy of *L. conspersus* with *L. nigromaculatus*. The holotype of this species survived the war and was rediscovered during our inspections.

The holotype is a large and robust lizard (Fig. 3). It is probably a male because the musculature associated with the hemipenes seems to be visible under the skin. The specimen is in relatively good condition, and only the left posterior margin of the mouth is damaged (Fig. 4). Also, the left side of the head is much more flattened than the right side. The teeth are highly worn but the posterior teeth are tricuspid.

The number of scales from the rostral to the occiput (the Hellmich index) is 13. The interparietal scale is large (slightly smaller than the parietals), subtriangular, with slightly concave right and slightly convex left margins; it contacts six scales. The frontal scale is single, approximately dumbbell-shaped, with a straight anterior margin and a posterior margin forming an obtuse angle. Five scales separate the rostral from the frontal, and seven scales separate it from the interparietal. The rostral contacts the nasal at a point. Nostrils are directed laterally on the right side and dorsolaterally on the left,

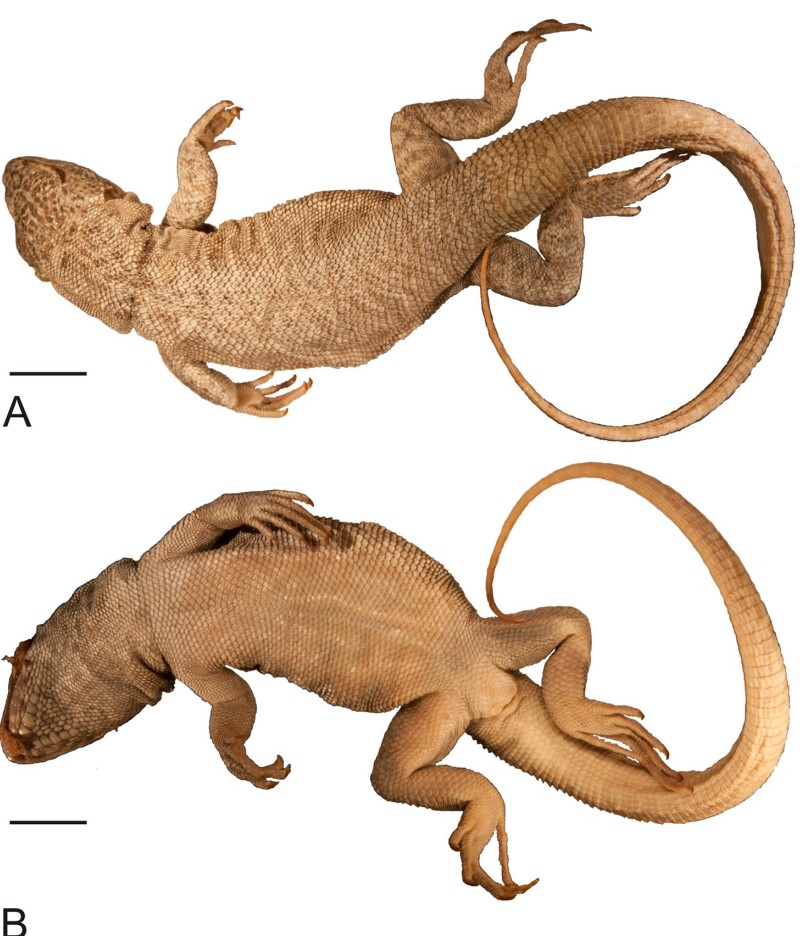

**Figure 3** Holotype of *Liolaemus conspersus Gravenhorst, 1838* (MNHUW 1321). (A) Dorsal view. (B) Ventral view. Scale bar = 1 cm. Photographs by Bartosz Borczyk.

probably due to distortion of the specimen. The nasal is separated from the canthal by one scale, and contacts eight scales. There are eight scales between the external nares. There are five flat and long supralabials; the fourth one is located below the eye and has an oblique posterior border. There are five loreals, one of which contacts the subocular. There is one row of seven lorilabials between the supralabials and loreals, four of which contact the subocular. Five infralabials are present; the second one contacts four scales, including two other infralabials (all counts of labial scales were made on the right side because of the damage on the left side of the specimen). There are five enlarged postmental scales (chinshields) on the right side; on the left, there are only four such scales, but the second one clearly originated by the fusion of two scales. The second postmentals are separated from each other by two gular scales.

The anteriormost gulars are slightly elongated and juxtaposed, while the others are wider than they are long and imbricate. The auricular scale is not differentiated. Temporal scales have varying shapes: some are subtriangular, some quadrangular, some pentagonal and some hexagonal; all are juxtaposed and unkeeled. There are approximately 73 dorsal

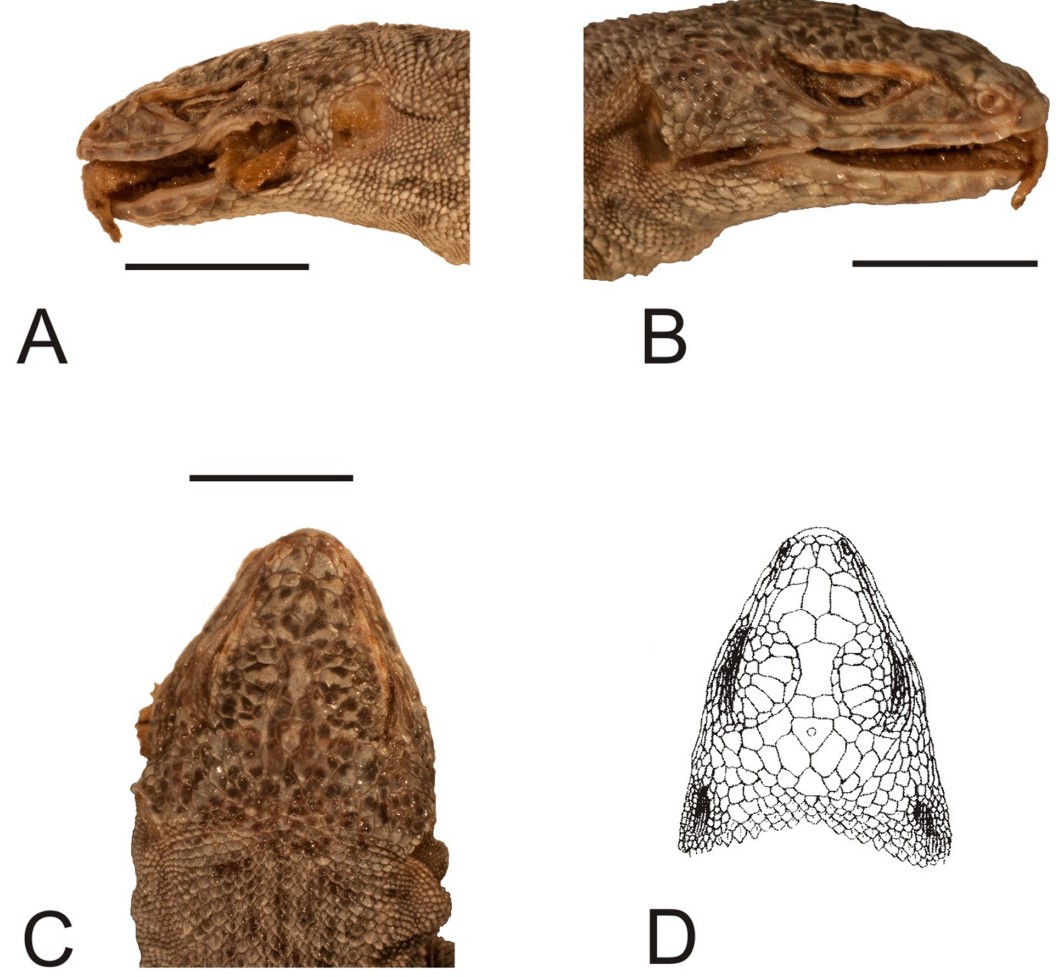

**Figure 4** **Head of the holotype of *Liolaemus conspersus Gravenhorst, 1838* (MNHUW 1321).** (A) Left lateral view. (B) Right lateral view. (C) Dorsal view. (D) Dorsal view as illustrated by *Gravenhorst (1838)*. Scale bar = 1 cm. Photographs by Bartosz Borczyk.

scales between the occiput and the anterior surface of the thighs. There are 115 ventral scales between the mental scale and the vent and 77 scales around the midbody. Most of the ventrals are quadrangular and juxtaposed or subimbricate, but some are pentagonal. The dorsal neck scales are rounded or lanceolate, strongly keeled, while the lateral neck scales are small and bead-like. The dorsal scales are larger than the dorsal neck scales but otherwise very similar (Fig. 5A). Interstitial granules are present between the dorsals. The postaxial surface of the forelimbs is covered by relatively large, rounded, imbricate scales. The arm scales are lightly keeled or unkeeled, while the forearm scales are strongly keeled. The preaxial surface of the arm is covered by much smaller, more rounded or bead-like, slightly imbricate scales. The numbers of subdigital lamellae on the left forelimb are as follows: I-11, II-16, III-22, IV-23, V-13; on the right forelimb, they are: I-12, II-18, III-22, IV-22, V-12. The dorsal thigh scales are rounded or lanceolate, imbricate and keeled. The lateral thigh scales are much smaller, rounded or bead-like. The shank scales are relatively large, rounded or lanceolate, very lightly keeled or

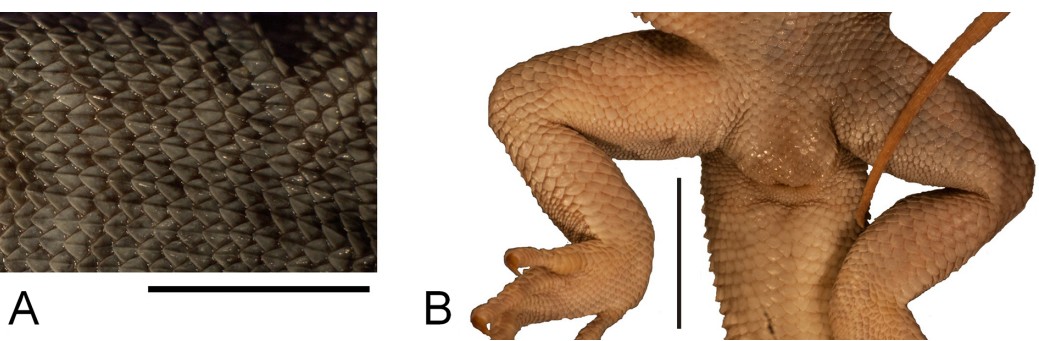

**Figure 5 Morphological details of the *Liolaemus conspersus* holotype (MNHUW 1321).** (A) Dorsal scales. Some of them are rounded, some are lanceolate, but all with strong keel. (B) Close-up of the cloacal region. No precloacal pores can be observed. Scale bar = 1 cm. Photographs by Bartosz Borczyk.

unkeeled. Postaxial scales on the proximal part of the shank are small, round and slightly imbricate. Both the dorsal and ventral foot scales are rounded or lanceolate and strongly keeled. No precloacal pores are visible (Fig. 5B). The number of subdigital lamellae on the left hindlimb are as follows: I-13, II-18, III-23, IV-31, V-20; on the right hindlimb, they are: I-14, II-19, III-24, IV-30, V-21.

The dorsal colouration is light grey, with the tail being slightly more yellowish. There is no difference in background colour between the vertebral and paravertebral fields. Vertebral, dorso- and ventrolateral stripes are absent. Numerous dark spots or short stripes are faintly visible on almost the entire dorsum and limbs; they are much more visible on the dorsal and lateral aspects of the head. The antehumeral spot cannot be unambiguously identified. Several darker longitudinal stripes are faintly visible on the throat. Apart from that, the ventral colouration is uniformly grey-yellowish.

*Liolaemus conspersus* does not have a patch of enlarged scales on the posterior surface of the thighs and its fourth supralabial scale is upturned, so its affiliation to the subgenus *Eulaemus*, and thus synonymy with *L. fitzingerii*, can be excluded (*Laurent, 1992*; *Lobo, Espinoza & Quinteros, 2010*). *Liolaemus conspersus* is the only species described by *Gravenhorst (1838)*, for which the subsequent synonimisation by *Boulenger (1885)* was later put into doubt (*Hellmich, 1934*), even though most authors accepted it (*Etheridge & Espinoza, 2000*; *Pincheira-Donoso, Scolaro & Sura, 2008*). *Liolaemus nigromaculatus*, its presumed senior synonym, and several closely related species, together belonging to the *nigromaculatus* species group, were redescribed and diagnosed in recent years (*Troncoso-Palacios & Garin, 2013*; *Troncoso-Palacios et al., 2015a*). While the diagnosis of the whole group is difficult to establish (*Troncoso-Palacios & Garin, 2013*), it seems that *L. conspersus* differs from most of its members in at least several characters. The nasal and rostral scales contact at a point in *L. conspersus*, which differentiates it from *L. nigromaculatus*, *L. atacamensis Müller & Hellmich, 1933*, *L. kuhlmanni Müller & Hellmich, 1933*, *L. silvai Ortiz, 1989* and *L. zapallarensis Müller & Hellmich, 1933*, in which these scales are separated. The second postmental scales are separated by two scales—a condition that is present only in some *L. atacamensis* individuals (6 out of 15, i.e. 40%, in the sample of *Troncoso-Palacios et al., 2015a*), while in others these scales

are separated by just one scale or are in contact. Interstitial granules are present between the dorsal scales in *L. conspersus*, as in some *L. nigromaculatus* and *L. atacamensis*, but unlike *L. silvai* and *L. zapallarensis*. *Liolaemus conspersus* has significantly more scales around the midbody (77) than any of the above-named lizards (48–62; *Troncoso-Palacios & Garin, 2013*; *Troncoso-Palacios et al., 2015a*). The holotype of *L. conspersus* is also larger than all the individuals of *L. atacamensis*, *L. kuhlmanni*, *L. silvai* and *L. nigromaculatus* examined by *Troncoso-Palacios & Garin (2013)* and *Troncoso-Palacios et al. (2015a)*, although only slightly larger in the case of the latter species. However, it should also be noted that the method and time of fixation and preservation of the specimen (which are impossible to compare between *L. conspersus* and specimens studied by *Troncoso-Palacios et al. (2015a)*) might have had an effect on its body measurements, especially the SVL (*Vervust, Van Dongen & Van Damme, 2009*).

It is also worth noting that the phylogeny and taxonomic content of the *nigromaculatus* group are not well established. Mitochondrial DNA analysis by *Troncoso-Palacios et al. (2015a)* indicates that all these species are closely related, but *Panzera et al. (2017)*, on the basis of analyses using 541 ultra-conserved elements and 44 protein-coding genes, suggested that *L. atacamensis* is only distantly related to the other named species, even though it is morphologically most similar to *L. nigromaculatus* (*Troncoso-Palacios & Garin, 2013*). Therefore, caution must be used when making taxonomic decisions based only on morphological data.

Members of the *nigromaculatus* group are also geographically separated from Cauquenes, from which the *L. conspersus* holotype is known. The nearest records are from Valparaíso, which is about 340 km north of Cauquenes (*Troncoso-Palacios et al., 2015a*). Other *Liolaemus* species known from Cauquenes (*L. chiliensis* (*Lesson, 1830*), *L. cyanogaster* (*Duméril & Bibron, 1837*), *L. lemniscatus*, *L. schroederi Müller & Hellmich, 1938*, *L. septentrionalis Pincheira-Donoso & Núñez, 2005*, *L. tenuis* (*Duméril & Bibron, 1837*)) differ from *L. conspersus* in size, number of scales around the midbody, shape of dorsal scales or a combination of the listed traits (*Pincheira-Donoso & Núñez, 2005*), so their conspecificity can be excluded. However, two species from the Maule Lagoon (located about 80 km east of Cauquenes) are very similar to *L. conspersus* in several morphometric and meristic characters. Both *Liolaemus buergeri Werner, 1907* and *Liolaemus flavipiceus* Cei & Videla, 2003 are of similar size to *L. conspersus* (maximum SVL of 96.2 and 95.8 mm, respectively; *Troncoso-Palacios et al., 2015b*). The number of dorsal scales in *L. buergeri* is quite variable—in the sample of *Troncoso-Palacios et al. (2015b)* it varied between 78 and 91 but *Medina, Avila & Morando (2013)* found values higher than 100. The number of ventral scales is between 111 and 125 (*Troncoso-Palacios et al., 2015b*). *Liolaemus flavipiceus* has 60–71 dorsal and 93–105 ventral scales (*Troncoso-Palacios et al., 2015b*). The number of midbody scales is 80–100 in *L. buergeri* and 68–83 in *L. flavipiceus* (*Garin et al., 2013*; *Troncoso-Palacios et al., 2015b*). However, this trait is geographically variable in *L. flavipiceus*—16 specimens collected by *Troncoso-Palacios et al. (2015b)* in Laguna del Maule had 68–78 scales but the single specimen from Paso Pehuenche (holotype) had 83 scales (*Garin et al., 2013*). In terms of the number of dorsal and ventral scales *L. conspersus* is more similar to *L. buergeri*

(although it has slightly fewer dorsals) than to *L. flavipiceus* but the converse is true for the number of midbody scales (*L. buergeri* has more than both *L. conspersus* and *L. flavipiceus*). It is also worth noting that *L. flavipiceus* is one of the few species in which males do not have precloacal pores (*Troncoso-Palacios et al., 2015b*). If the holotype of *L. conspersus* is indeed a male, this would suggest its conspecificity with *L. flavipiceus*. However, at present it is difficult to decide about the potential synonymy of *L. conspersus* with one of these species; further studies are needed to determine whether this is possible to assess. This is important because the name *Liolaemus conspersus* would have priority over both *L. buergeri* and *L. flavipiceus* (see also *Discussion*). At present, we regard it as a *species inquirenda*, that is, a species of doubtful identity, needing further investigation (*International Commission on Zoological Nomenclature, 1999*).

*Liolaemus hieroglyphicus* Gravenhorst, 1838

**Original name.** *Liolaemus hieroglyphicus* Gravenhorst, 1838: 732.

**Type specimen.** A single specimen (MNHUW 1322, probably female, SVL 54.0 mm) collected by F.S. Scholtz before 1838 in 'Cauquenes' (in the current Maule region, Chile). This specimen still exists in Wrocław.

**Present name.** *Liolaemus hieroglyphicus* Gravenhorst, 1838.

**Remarks.** *Gravenhorst (1838)* described this species on the basis of a single individual from Cauquenes in Chile. It was synonymised with *Ptychodeira signifera* (*Duméril & Bibron, 1837*) (currently *Liolaemus signifer*) by *Fitzinger (1843)*, with *L. olivaceus* (currently regarded as a synonym of *L. chiliensis*; *Donoso-Barros, 1966*) by *Tschudi (1845)* and with *L. lemniscatus* by *Boulenger (1885)*. Unfortunately, this specimen was not illustrated by *Gravenhorst (1838)*. However, we think that an unlabelled specimen found in the collection is the missing holotype. Our assertion is based on the following reasons: (1) *Gravenhorst (1838)* stated that 5/6 of the holotype tail is regenerated and a comparable part of the tail is regenerated in the rediscovered specimen; (2) there is an indentation around the neck, which was also noted by *Gravenhorst (1838)* in *L. hieroglyphicus* (also in *L. unicolor*, which is, however, much larger than the specimen described herein); (3) the individual is a similar size and colour to those reported originally by *Gravenhorst (1838)* and no other specimen in the collection exhibits similar traits.

Immediately after *L. hieroglyphicus*, *Gravenhorst (1838)* also described a 'variety', which he apparently regarded as an intermediate between *L. hieroglyphicus* and *L. lemniscatus*. However, it is unclear whether he intended this form to be a variety of the former species (this is the position taken by *Tschudi (1845)*) or regarded it as a taxon of a yet unclear taxonomic position. It was based on a smaller individual than the type of *L. hieroglyphicus*. We were not able to locate this specimen. Regardless of these uncertainties, the surviving specimen of *L. hieroglyphicus* is the only type specimen of the type variety and thus can be regarded as the holotype of that species.

The rediscovered holotype is in very good condition (Figs. 6 and 7). There are no visible damages, except the indentation around the neck, which was already present at the

 

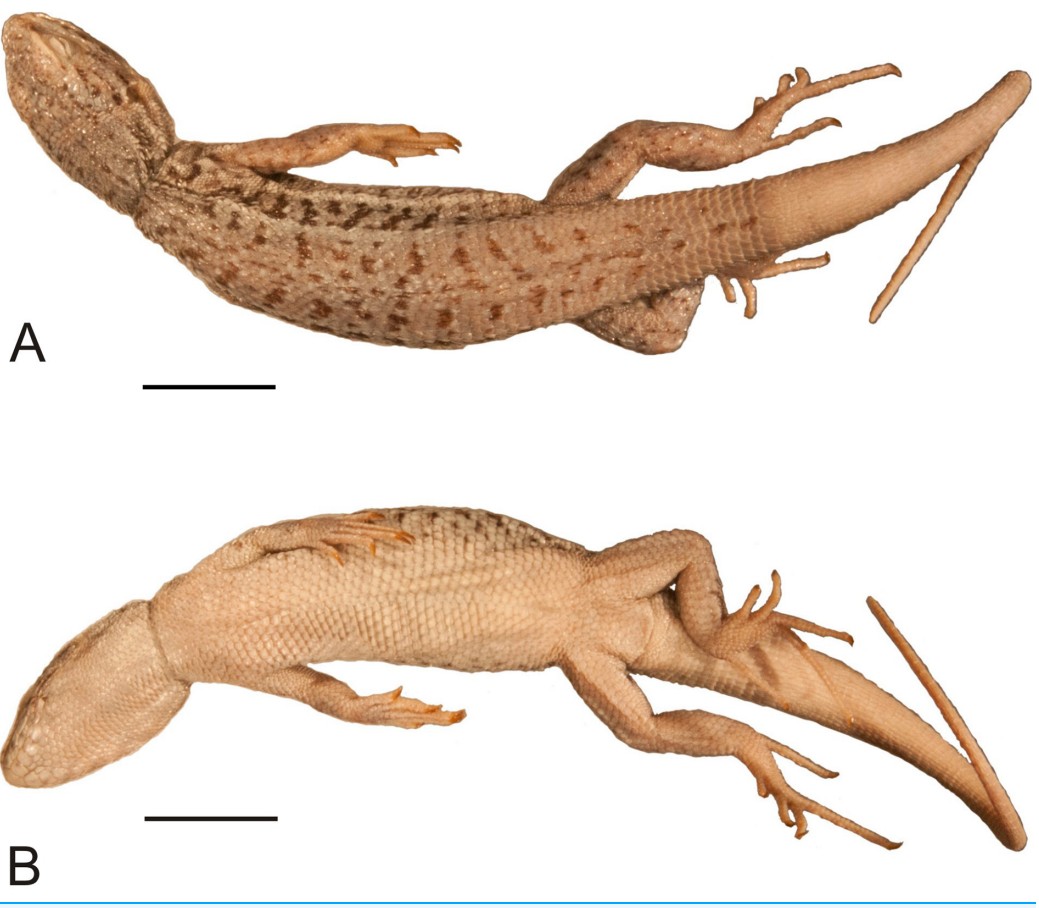

**Figure 6 Holotype of *Liolaemus hieroglyphicus Gravenhorst, 1838* (MNHUW 1322).** (A) Dorsal view. (B) Ventral view. Scale bar = 1 cm. Photographs by Bartosz Borczyk.

time of Gravenhorst's work; even the colour pattern is apparently preserved close to the original.

The Hellmich index is 11. The interparietal scale is hexagonal but with rounded margins, contacts six scales and is much smaller than the parietals. The interparietal is separated by seven scales (at the midline) from the rostral. The frontal scale is hexagonal but with a nearly straight anterior margin, and is separated from the rostral by five scales. The rostral and nasal are in broad contact. The nasal is separated from the canthal by one scale. There are two pairs of internasals, the first one is in medial contact, the second pair is separated by one scale. Posteriorly, they contact two large, hexagonal frontonasals, also separated medially by one scale. There are five supralabials on both the left and right sides, the fourth one is located below the eye and has an upturned posterior margin. One row of lorilabials separates the supralabials from the loreals. The subocular contacts four lorilabials on the right side but only two on the left side. The loreal region is concave, and contains four loreals on the left and four on the right. The temporal scales are polygonal, some with a rounded posterior margin, juxtaposed or slightly imbricate, not keeled (only a few upper temporals are

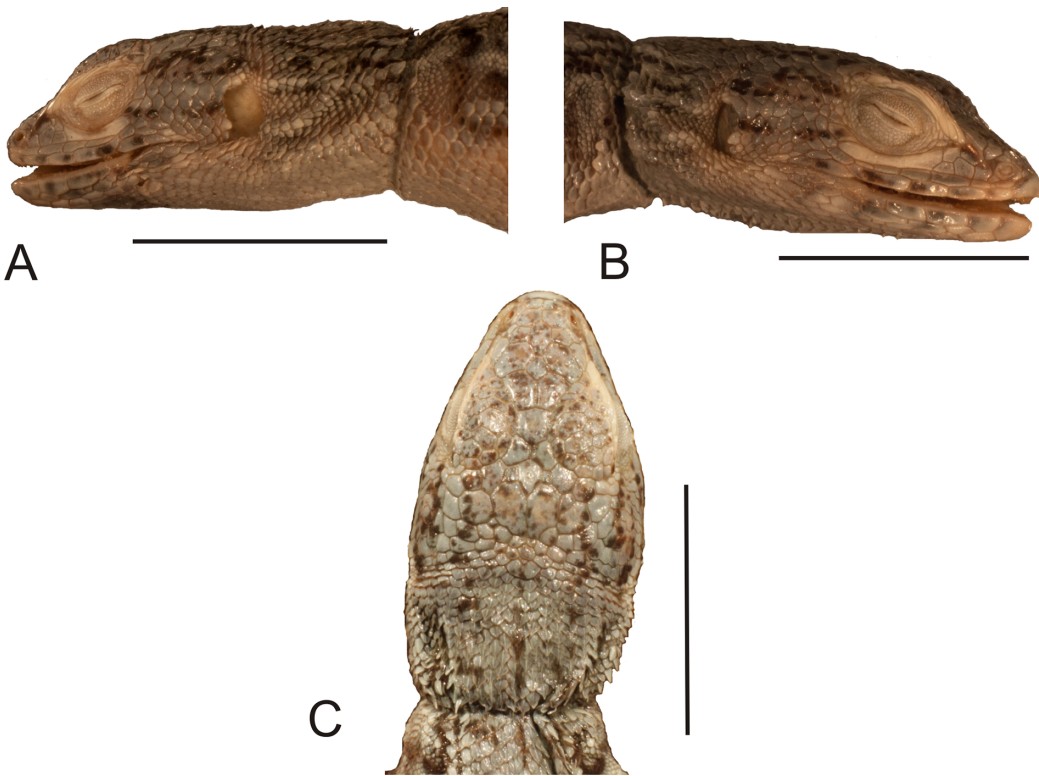

**Figure 7 Head of the holotype of *Liolaemus hieroglyphicus Gravenhorst, 1838*** (MNHUW 1322). (A) Left lateral view. (B) Right lateral view. (C) Dorsal view. Scale bar = 1 cm. Photographs by Bartosz Borczyk.

very lightly keeled). The auricular scale is present, and its ventral border does not reach the auditory meatus midline. There are three enlarged supraoculars, with the first one being the largest. Seven supraciliaries are present. Four infralabials are present on the right side and five on the left side; the second one contacts four scales, including two other infralabials. There are four pairs of enlarged postmental scales (chinshields); the first pair is in contact, and the second pair is separated by two scales. Most gulars are oval, slightly imbricate.

The nuchal and dorsal scales are imbricate, keeled and lanceolate with mucrons (some rounded without mucrons; Fig. 8A). Only the tail scales lack keel. There are 52 dorsal scales, 84 ventral scales and 56 scales around the midbody. The ventral scales are round, juxtaposed or only minimally imbricate. No precloacal pores can be observed (Fig. 8B). The lateral scales are similar to the ventral scales, and they are only much smaller around the limbs, where they are round or bead-like. The forelimb scales are round or lanceolate, imbricate, and only some on the postaxial surface are lightly keeled. The posterior surface of the arms is covered by round, granular, slightly imbricate scales. The numbers of subdigital lamellae on the left forelimb are as follows: I-8, II-12, III-16, IV-16, V-9; on the right forelimb, they are: I-7, II-12, III-15, IV-15, V-8. Scales on the hindlimbs are very similar to those on the venter; only on the foot they are much smaller and quadrangular rather than round. The numbers of subdigital lamellae on the

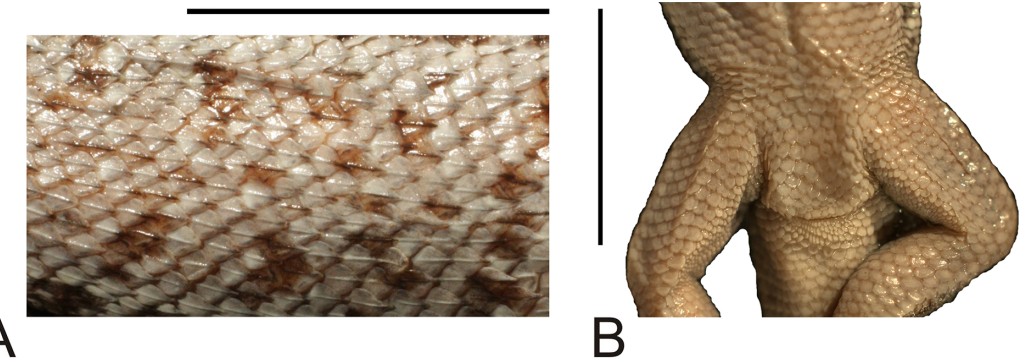

**Figure 8** **Morphological details of the *Liolaemus hieroglyphicus* holotype (MNHUW 1322).** (A) Dorsal scales. (B) Close-up of the cloacal region. No precloacal pores can be observed. Scale bar = 1 cm. Photographs by Bartosz Borczyk.

left hindlimb are as follows: I-8, II-13, III-17, IV-22, V-14; on the right hindlimb, they are: I-8, II-14, III-18, IV-22, V-13.

The dorsal background colouration is almost uniformly grey. There are numerous short (usually two to three scales long) brown stripes extending from the occiput to the preserved part of the tail, resembling a dotted line running along the spine. Similar stripes are also present in the temporal region. Numerous small brown spots occur on almost the entire pileus (usually several on a single scale), the supralabials and the infralabials. The paravertebral fields are covered by many brown spots, usually one to two scales long and two to four scales wide. Whitish dorso- and ventrolateral lines are present. Between them, the lateral fields are also covered by brown-black spots, which are less numerous but larger than the paravertebral spots. The whole ventral body part is uniformly whitish.

*Liolaemus hieroglyphicus* does not have a patch of enlarged scales on the posterior surface of the thighs and its fourth supralabial scale is upturned, so its affiliation to the subgenus *Eulaemus*, and thus synonymy with *L. signifer*, can be excluded (*Laurent, 1992*; *Lobo, Espinoza & Quinteros, 2010*). It is also not synonymous with *L. chiliensis*, as it has significantly more scales around the midbody (56 in *L. hieroglyphicus*, 31–33 in *L. chiliensis*; *Pincheira-Donoso & Núñez, 2005*). The holotype of *L. hieroglyphicus* shows many similarities to *L. lemniscatus*, with which it was synonymised (*Boulenger, 1885*). It is of similar size (SVL), has similar distance between the axilla and groin and a similar number of ventral scales and scales around the midbody (*Martinez et al., 2011*; *Quinteros, 2012*). It shares several scalation traits with the *L. lemniscatus* holotype (*Borczyk, 2013*), such as a hexagonal interparietal, much smaller than the parietals; the presence of three enlarged supraorbitals, of which the first one is the largest; contact between the nasal and rostral; separation of the nasal and canthal by one scale; keeled, mucronate, imbricate dorsal scales. However, these two specimens differ significantly in the scalation of the anterior part of the pileus. In *L. lemniscatus*, there are no single median scales, while in *L. hieroglyphicus*, two such scales are present, separating the second pair of internasals and both frontonasals. Also, the lateral neck scales in the former species are lanceolate,

keeled and imbricate, while in the latter they are granular and unkeeled, being more similar to those in *L. pseudolemniscatus* (*Troncoso-Palacios, 2011*). Temporal scales are markedly keeled in *L. lemniscatus* (*Quinteros, 2012*), while in *L. hieroglyphicus*, most scales lack keel, and only a few upper temporals are very lightly keeled. *Liolaemus hieroglyphicus* has fewer subdigital lamellae on the fourth toe than the *L. lemniscatus* specimens listed by *Martinez et al. (2011)*, but more than the holotype of that species (*Borczyk, 2013*). *Liolaemus hieroglyphicus* differs from *L. abdalai* (very similar species to *L. lemniscatus*) by its larger size (the maximum SVL in *L. abdalai* is 50.2 mm according to *Quinteros, 2012*). The auricular scale in *L. hieroglyphicus* is smaller than in members of the *L. lemniscatus* group, not reaching the auditory meatus midline (*Quinteros, 2013*). *Liolaemus hieroglyphicus* differs from *L. septentrionalis* by having fewer scales around the midbody (56 in *L. hieroglyphicus*, 63–73 in the latter), more loreal scales (four and two to three, respectively), fewer subdigital lamellae on the fourth toe (22 and 25–28, respectively) and in the shape of the dorsal scales, which are more rounded and do not have a mucron in *L. septentrionalis* (*Pincheira-Donoso & Núñez, 2005*). *Liolaemus hieroglyphicus* is similar to members of the *Liolaemus gravenhorstii* group (composed of *Liolaemus gravenhorstii* (*Gray, 1845*), *L. schroederi* and *L. cyanogaster* (*Quinteros, 2013*), but sometimes also including other species such as *L. chiliensis* (*Portelli & Quinteros, 2018*)). However, it differs from *L. gravenhorstii* and *L. cyanogaster* by having more scales around the midbody (40–43 in *L. gravenhorstii* and 45–50 in *L. cyanogaster*; *Pincheira-Donoso & Núñez, 2005*), and in the shape of the lateral neck scales, which are unkeeled and usually rounded in *L. hieroglyphicus*, while in *L. gravenhorstii* they are large, lanceolate, imbricate and strongly keeled and in *L. cyanogaster*, they are small, triangular, imbricate and moderately keeled (*Pincheira-Donoso & Núñez, 2005*). It further differs from *L. cyanogaster* in the shape of the ventral scales, which are imbricate and similar to the dorsals in the latter (*Pincheira-Donoso & Núñez, 2005*), while in *L. hieroglyphicus* they are rounded or subquadrangular and only minimally imbricate, as well as in the number of subdigital lamellae on the fourth toe (22 in *L. hieroglyphicus*, 25–30 in *L. cyanogaster*; *Pincheira-Donoso & Núñez, 2005*). *Liolaemus schroederi* also has slightly fewer dorsal scales (46–53), but more subdigital lamellae on the fourth toe (24–28). The shape of the ventral scales is very variable in this species—they can be triangular, subtriangular or rounded, as in *L. hieroglyphicus*. The lateral neck scales are markedly keeled in *L. schroederi* (*Pincheira-Donoso & Núñez, 2005*), while in *L. hieroglyphicus* only the scales located posterodorsally to the auditory meatus have keel. *Liolaemus gravenhorstii* is also geographically separated from *L. hieroglyphicus*, being known primarily from the Metropolitan Region (*Pincheira-Donoso & Núñez, 2005*).

*Liolaemus hieroglyphicus* is not synonymous with *L. signifer*, *L. chiliensis* or *L. lemniscatus*. It is similar to *L. cyanogaster* and *L. schroederi* (particularly the latter), but at present cannot be unambiguously synonymised with any of these. We regard it as a *species inquirenda*, that is, a species of doubtful identity, needing further investigation (*International Commission on Zoological Nomenclature, 1999*).

*Liolaemus lemniscatus* Gravenhorst, 1838

**Original name.** *Liolaemus lemniscatus* Gravenhorst, 1838: 731.

**Type specimen.** A single specimen (MNHUW Re 0027, female) collected by F.S. Scholtz in 'Valparaíso', Chile.

**Present name.** *Liolaemus lemniscatus* Gravenhorst, 1838.

**Remarks.** The holotype of this species survived the war and was recently redescribed (Borczyk, 2013).

*Liolaemus lineatus* Gravenhorst, 1838

**Original name.** *Liolaemus lineatus* Gravenhorst, 1838: 723.

**Type specimens.** Four specimens (see also the Remarks section below) of undetermined sex, collected by F.S. Scholtz in 'Valparaíso', Chile. Two specimens (MNHUW 1323a, SVL 89.4 mm; MNHUW 1323b, SVL 54.3 mm) still exist in Wrocław, and the two others are probably lost.

**Present name.** *Liolaemus nitidus* (Wiegmann, 1834).

**Remarks.** Gravenhorst (1838) distinguished four varieties of this species (one main and three subordinate ones—'Hauptart' and 'Abarten' in German). Although the exact number of specimens he had studied is not stated in his article, this suggests that he had at least four specimens in his collection; however, only three individuals are illustrated—one specimen each of the main variety, the second and third varieties. Unfortunately, the first variety was not illustrated. However, in a museum catalogue from before 1907, four specimens are listed, though later note updates it to only two specimens (Fig. 9A). The second catalogue that survived the war (made in 1907; Fig. 9B) lists only three specimens. It is thus possible that some of these lizards were lost even before the war; it cannot be excluded that they had been sent to other institutions but this is not indicated.

All specimens of *Liolaemus lineatus* were collected in Valparaíso, Chile. The first and second varieties were synonymised with *L. olivaceus* (Wiegmann, 1834) (currently considered a synonym of *L. chiliensis* (Lesson, 1830); Pincheira-Donoso & Núñez, 2005) by Fitzinger (1843), while the third variety was synonymised with *L. chiliensis* (he did not mention the main variety). Boulenger (1885) synonymised this species (all four varieties) with *L. nitidus*. *Liolaemus lineatus* Gravenhorst, 1838 should not be confused with its younger homonym *Liolaemus lineatus* Gray, 1845, a species currently regarded as synonymous with *Liolaemus nigroviridis* Müller & Hellmich, 1932 (Núñez, 2004).

Two individuals, belonging to the second and third varieties, were rediscovered but the specimen representing the unillustrated first variety and a specimen of the main variety are probably lost. Both rediscovered specimens were desiccated and in very poor condition; however, rehydration in 0.5% $Na_3PO_4$ improved their condition and allowed us to describe many taxonomically informative characters.

**Figure 9 Excerpts from two surviving pre-World War II catalogues of herpetological specimens in the Museum of Natural History in Wrocław.** (A) Catalogue from before 1907. (B) Catalogue from 1907.

The type specimen of the second variety (MNHUW 1323a) is a relatively large and robust lizard with a stout appearance (Figs. 10 and 11). The specimen is damaged on the right side of the venter, where scales are missing. The teeth are highly worn but the anterior teeth appear to be conical, while the posterior teeth are multicusped. At least one tooth is visible on the left pterygoid and at least one is visible on the right pterygoid. The Hellmich index is 13. The interparietal scale is pentagonal, elongated posteriorly, smaller than the parietals. It contacts six scales and is separated by nine scales from the rostral. The frontal scale is single, large, pentagonal, with almost straight anterior and right margins and a slightly concave left margin; it is separated from the rostral by seven scales. The internarial region is fragmented asymmetrically into six scales. The nasal scale contacts seven scales; it is in broad contact with the rostral and is separated by one scale from the canthal. A roughly rhomboidal scale is located between the prefrontals and frontonasals. There is also a single median scale between the anterior parts of the frontonasals. There are three enlarged supraoculars, with the first one being the largest and the second one being—the smallest. Five supralabials are present on both sides, with the fourth one located below the eye and having an upturned posterior border. The loreal region is concave, containing six scales (eight on the left side), two of which contact the subocular (three on the left side); it is separated from the supralabials by one row of lorilabials (five present on both the left and right sides). Five infralabials are visible on the right side but only four are visible on the left. There are three pairs of enlarged postmental scales (chinshields); the first one is in contact, and the second one is medially separated by two scales. The temporal scales are round or lanceolate, keeled and imbricate. A small patch of much smaller, oval and juxtaposed scales covers the

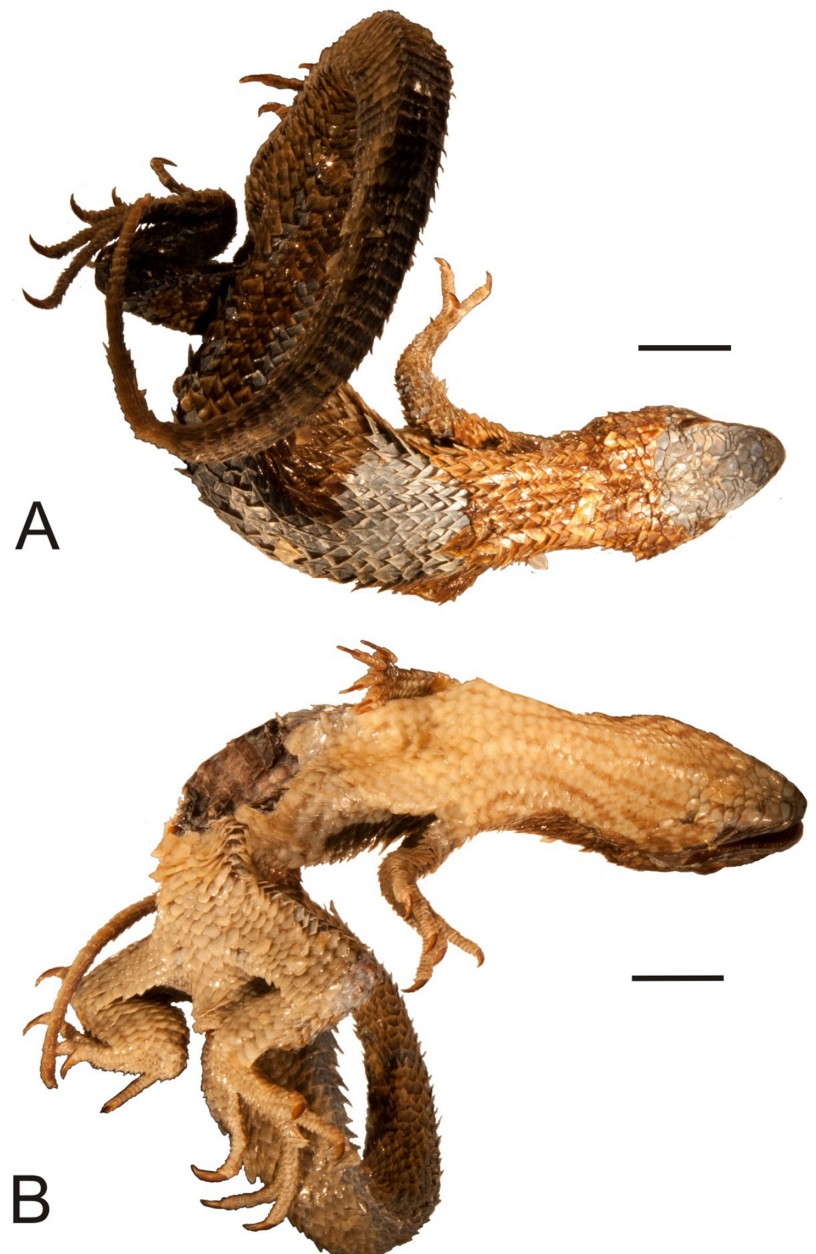

**Figure 10 Lectotype of *Liolaemus lineatus Gravenhorst, 1838* (MNHUW 1323a).** (A) Dorsal view. (B) Ventral view. Scale bar = 1 cm. Photographs by Bartosz Borczyk.

posterior border of the auditory meatus. The number of scales between the posterior border of the auditory meatus and the shoulder is 27 on the left side and 25 on the right side.

The dorsal scales are large, imbricate, strongly keeled and mucronate. There are 33 dorsal scales, 69 ventral scales and 46 scales around the midbody. The tail scales form numerous annuli, which are particularly visible from the ventral side. The ventral tail scales have smaller mucrons. The ventral scales (including the gulars) are also relatively

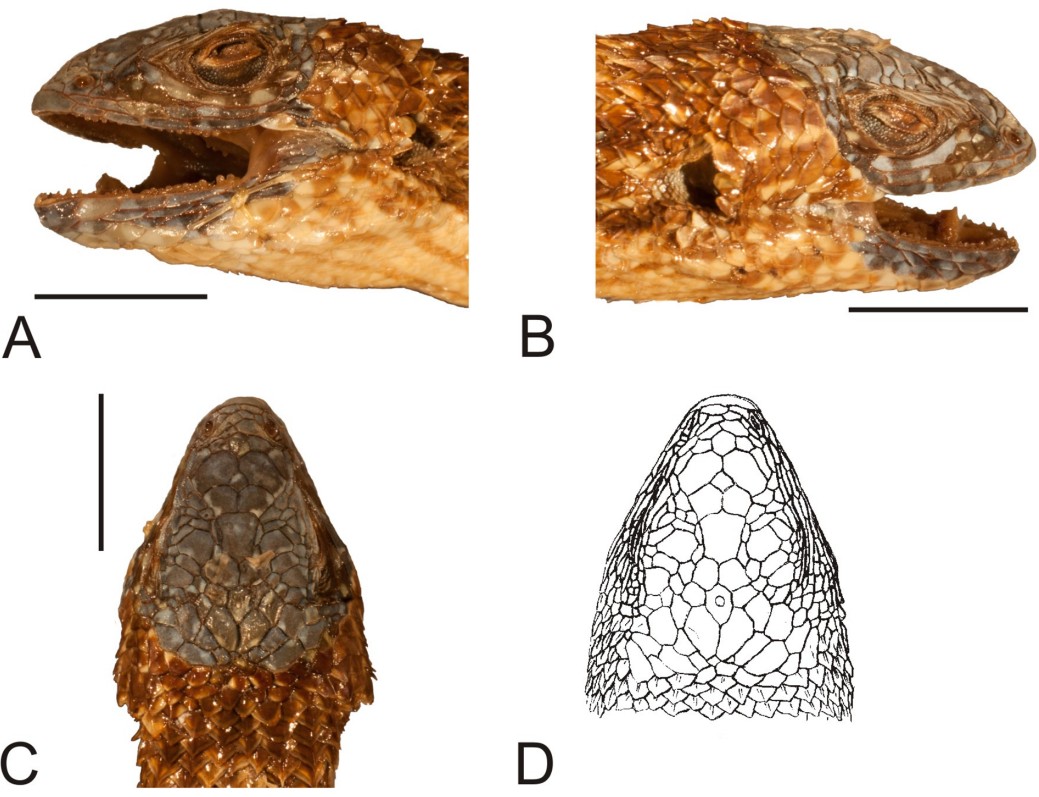

**Figure 11 Head of the lectotype of *Liolaemus lineatus Gravenhorst, 1838* (MNHUW 1323a).** (A) Left lateral view. (B) Right lateral view. (C) Dorsal view. (D) Dorsal view as illustrated by *Gravenhorst (1838)*. Scale bar = 1 cm. Photographs by Bartosz Borczyk.

large, lanceolate or rounded (particularly the gulars), imbricate and not keeled. There are 29 gular scales between auditory meatus. The posterior parts of the thighs are covered by much smaller, oval, juxtaposed scales. No precloacal pores are visible. The numbers of subdigital lamellae on the left forelimb are as follows: I-8, II-15, III-17, IV-21, V-12; on the right forelimb, they are: I-9, II-15, III-19, IV-20, V-11; on the left hindlimb, they are: I-13, II-17, III-21, IV-25, V-14; on the right hindlimb, they are: I-10, II-15, III-18, IV-25, V-14.

The colour of the head and dorsum is blue-grey. Numerous darker spots are visible on the lateral side of the head; they cover the background colour on the pileus. The ventral body part is almost uniformly white-yellow, with the throat covered by several faintly visible darker longitudinal stripes.

The type specimen of the third variety (MNHUW 1323b) is much smaller than MNHUW 1323a (Figs. 12 and 13). It has minor damage on the venter. The Hellmich index is 13. The interparietal is hexagonal, expanded posteriorly and smaller than the parietals. It contacts six scales. The frontal scale is paired; the right one is hexagonal and elongated, while the left one is shorter and has more rounded margins. Five scales separate the rostral and frontal scales and seven separate the rostral and interparietal. The internarial region is fragmented into six scales, with one scale at the midline. Posteriorly to the postrostrals, there are three scales, arranged in a midline row and

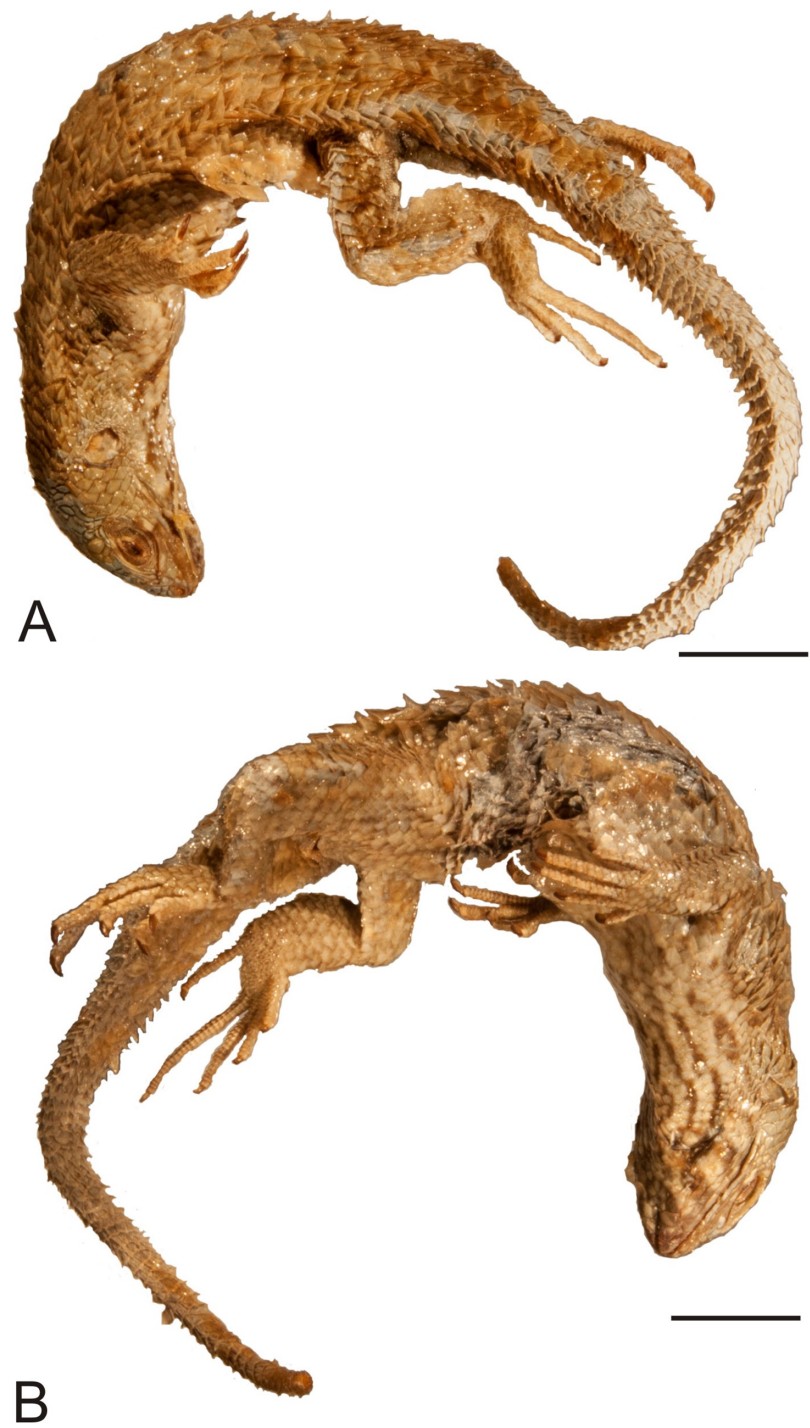

**Figure 12  Paralectotype of *Liolaemus lineatus Gravenhorst, 1838* (MNHUW 1323b).** (A) Dorsal view. (B) Ventral view. Scale bar = 1 cm. Photographs by Bartosz Borczyk.

separating the frontonasals from each other. The nasal scale is pentagonal, contacts six scales and is separated both from the canthal and the rostral by one scale. There are three enlarged supraoculars, with the first one being the largest. The loreal region is slightly

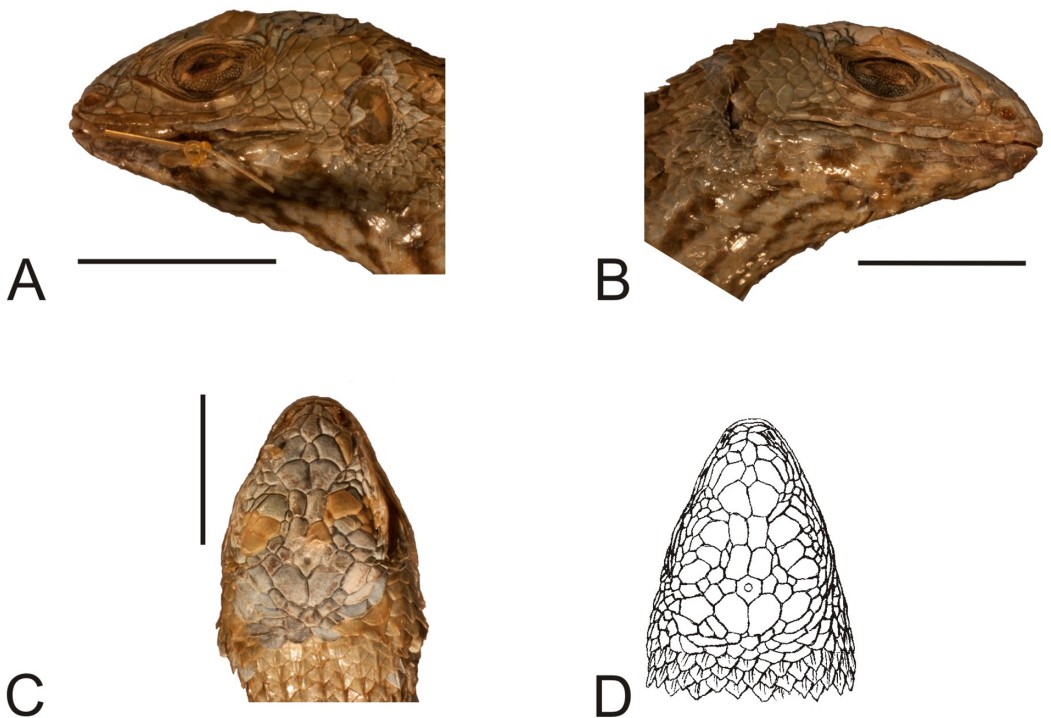

**Figure 13 Head of the paralectotype of *Liolaemus lineatus Gravenhorst, 1838* (MNHUW 1323b).** (A) Left lateral view. (B) Right lateral view. (C) Dorsal view. (D) Dorsal view as illustrated by *Gravenhorst (1838)*. Scale bar = 1 cm. Photographs by Bartosz Borczyk.

concave and contains six scales, three of which contact the subocular. Five supralabials are present on both the left and right sides; the fourth one is located below the eye and has an upturned posterior margin. One row of lorilabials separates them from the loreals (there are five lorilabials on each side, two contacting the subocular). There are five infralabials on both sides. Three pairs of enlarged postmental scales (chinshields) are present, the first pair in contact, and the second pair separated by two scales. The temporal scales are imbricate, rounded or lanceolate and keeled (only the smallest ones lack keel). The ventral and posterior borders of the auditory meatus are covered by very small, lanceolate, unkeeled scales.

The dorsal and ventral scalation is very similar to the second variety—dorsal scales are imbricate, lanceolate (only the anteriormost nuchal scales are rounded), keeled and mucronate (although the mucrons are slightly smaller), while the ventrals are large, flat, rounded (gulars and ventral neck scales) or lanceolate, imbricate and unkeeled. There are 33 dorsal scales, 64 ventral scales and 43 scales around the midbody. A total of 30 gulars are present between the auditory meatus. No precloacal pores can be observed. The number of scales between the posterior border of the auditory meatus and the shoulder is 16 on the left side and 18 on the right side. The numbers of subdigital lamellae on the left forelimb are as follows: I-9, II-15, III-18, IV-16, V-12; on the right forelimb, they are: I-9, II-14, III-18, IV-16, V-10; on the left hindlimb, they are: I-10, II-16, III-20, IV-26, V-14; on the right hindlimb, they are: I-11, II-15, III-21, IV-25, V-14.

The colouration is very similar to the second variety. The dark stripes on the throat are much more visible in this specimen than in MNHUW 1323a. On the other hand, the spots on the lateral part of the head are better preserved in the type specimen of the second variety; however, this may be an artefact of preservation.

Relationships of *Liolaemus nitidus* have long been difficult to establish (*Pincheira-Donoso & Núñez, 2005*). Most recent morphological studies agree on its affinities to *L. robertmertensi* and *L. chiliensis* (*Lobo, 2001*, *2005*; *Pincheira-Donoso & Núñez, 2005*; *Quinteros, 2013*). However, molecular analyses indicate completely different phylogenetic position, particularly its close relationship to *L. monticola* (*Troncoso-Palacios et al., 2015a*; *Panzera et al., 2017*; *Torres-Pérez et al., 2017*). It was always only distantly related to both *L. robertmertensi* and *L. chiliensis*, which were also distantly related to each other, in mitochondrial DNA analyses conducted by *Troncoso-Palacios et al. (2015a)* and combined morphological and nuclear DNA analyses performed by *Portelli & Quinteros (2018)*. Thus, it is difficult to evaluate potential synonymy based only on morphological characters. The type specimen of the second variety is indeed similar to *L. nitidus* in its large size, robustness, presence of lanceolate, keeled and mucronate dorsal scales and imbricate and keeled temporal scales. Moreover, the SVL, the number of scales around the midbody and the subdigital lamellae on the fourth toe are within the range observed in *L. nitidus* (*Pincheira-Donoso & Núñez, 2005*). We failed to note any character differentiating these two forms, so, currently, there seems to be no reason to disprove the synonymy of the second variety with *L. nitidus* (*Boulenger, 1885*). However, the third variety differs from the second one (and from *L. nitidus*; *Pincheira-Donoso & Núñez, 2005*) in several characters that have been regarded as taxonomically informative such as the presence of two frontal scales rather than one (also, with a different shape than in *L. nitidus*) and the contact between the nasal and rostral scales. In all 16 specimens of *L. nitidus* sampled by Troncoso-Palacios, the nasal and rostral scales are in contact; this trait is also apparently fixed in *L. chiliensis* (J. Troncoso-Palacios, 2018, personal communication—see 'peer review history'). The synonymy of the third variety with *L. nitidus* is more problematic. It is more similar to *L. chiliensis*, a species with which it was synonymised by *Fitzinger (1843)*, by having a subdivided frontal scale. However, it differs from that species and from *L. nitidus* by the presence of contact between the rostral and nasal scales (*Pincheira-Donoso & Núñez, 2005*).

It cannot be excluded that the type specimen of the third variety is a *Liolaemus nitidus* specimen with several scale anomalies but this is very difficult to test without molecular data. In light of the differences described above, the conspecificity of the type specimens of the second and the third varieties is problematic. According to *Gravenhorst's (1838)* drawings, the second variety is much more similar to the type variety than to the third one in all traits that could be evaluated (e.g. the presence of the single frontal scale, contact between the rostral and nasal scales, the presence of a small rhomboidal scale between the prefrontals and frontonasals). Therefore, we designate this specimen (MNHUW 1323a) as the lectotype of *L. lineatus*. Because this name has been considered synonymous with *L. nitidus* since at least 1885 (*Boulenger, 1885*), doing so would best serve

the nomenclatural stability. The type specimen of the third variety thus becomes the paralectotype, at least until its taxonomic status (potential conspecificity with the lectotype) is resolved.

## Missing types

Reptilia *Laurenti, 1768*
Squamata *Oppel, 1811*
Liolaemidae *Frost & Etheridge, 1989*

*Liolaemus marmoratus Gravenhorst, 1838*

**Original name.** *Liolaemus marmoratus Gravenhorst, 1838*: 729.

**Type specimen.** A single specimen of unspecified sex collected by F.S. Scholtz in 'Cauquenes' (in the current Maule region), Chile. We did not find this specimen and consider it to be lost.
**Present name.** *Liolaemus nitidus* (*Wiegmann, 1834*).

**Remarks.** *Liolaemus marmoratus* was synonymised with *L. nitidus* by *Boulenger (1885)*. The name *Liolaemus marmoratus* was also coined by *Burmeister (1861)* for the species of an Argentinean liolaemid. However, as it is homonymous with that coined by *Gravenhorst (1838)*, it was later replaced by *Liolaemus pseudoanomalus Cei, 1981*. We were not able to find the type of *L. marmoratus* and consider it to be most probably lost.

*Liolaemus unicolor Gravenhorst, 1838*

**Original name.** *Liolaemus unicolor Gravenhorst, 1838*: 728.

**Type specimen.** A single specimen of unspecified sex, collected by F.S. Scholtz in 'a small town in Cauquenes, about 20 German miles south of St. Jago, at the foot of the Cordilleras, where hot springs are present' (about 150 km south of Santiago, Chile). This specimen is probably lost.

**Present name.** *Liolaemus nitidus* (*Wiegmann, 1834*).

**Remarks.** This species was synonymised with *L. nitidus* by *Boulenger (1885)*. Unfortunately, it was not illustrated. The morphology of one very poorly preserved *Liolaemus* specimen (MNHUW uncatalogued) is consistent with *Gravenhorst's (1838)* description of *L. unicolor* but no unambiguous identification can be made. Therefore, this name must be considered a *nomen dubium*.

Reptilia *Laurenti, 1768*
Squamata *Oppel, 1811*
Leiocephalidae *Frost & Etheridge, 1989*

*Leiocephalus schreibersii* (*Gravenhorst, 1838*)

**Original name.** *Pristinotus schreibersii* *Gravenhorst, 1838*: 739.

**Type specimen.** A single specimen of unspecified sex, collected in 'St. Domingo' (Santo Domingo, Dominican Republic) by natural history dealer Ludwig Parreys (1796–1879) from Vienna. We did not find this specimen and consider it to be lost.

**Present name.** *Leiocephalus schreibersii* (*Gravenhorst, 1838*).

**Remarks.** The iguanian lizard *Leiocephalus schreibersii* was originally described by Gravenhorst under the name *Pristinotus schreibersii* on the basis of a single individual collected by Parreys in Santo Domingo, Dominican Republic (*Gravenhorst, 1838*). The holotype has long been thought to be lost (*Pregill, 1992*) and we were also unable to locate it in the collection. Recently, a neotype (SMF 26228) has been designated for this species (*Köhler, Rodríguez Bobadilla & Hedges, 2016*).

Reptilia *Laurenti, 1768*
Squamata *Oppel, 1811*
Scincidae *Oppel, 1811*

*Chalcides viridanus* (*Gravenhorst, 1851*)

**Original name.** *Gongylus viridanus* *Gravenhorst, 1851*: 348.

**Type specimens.** Three specimens collected on 'Teneriffa' (Tenerife) by a natural history dealer, Bescke; there are several naturalists with this name, so it is unclear whether one of them collected lizards for Gravenhorst. We were not able to locate any of these specimens and consider them to be lost.

**Present name.** *Chalcides viridanus* (*Gravenhorst, 1851*).

**Remarks.** This species of scincine skink was described by *Gravenhorst (1851)* as *G. viridanus* on the basis of three individuals collected on Tenerife by Bescke. It also occurs on the islands of El Hierro and Gomera and a few smaller islets (*Miras, Pérez-Mellado & Martínez-Solano, 2009*). Populations inhabiting these three large islands form three separate clades, with lizards from El Hierro and Gomera probably being sister groups (*Brown & Pestano, 1998*). The population from Tenerife is not homogenous and exhibits substantial divergence in mitochondrial DNA sequences, resulting in several geographical clusters (*Brown, Campos-Delgado & Pestano, 2000*; *Brown, Woods & Thorpe, 2017*). However, nuclear DNA shows only shallow divergences and a weak geographical pattern (*Brown, Woods & Thorpe, 2017*). Recently, J. Mateo (cited by *Miras, Pérez-Mellado & Martínez-Solano (2009)*) suggested that *Chalcides viridanus* probably represents a species complex and should possibly be split. Thus, a detailed description of the syntypes of this species would be useful for potential future works on its taxonomy. Unfortunately, we were unable to locate these specimens in the collection. Two well-preserved individuals collected on Tenerife by Zimmer in 1906 have been rediscovered (MNHUW

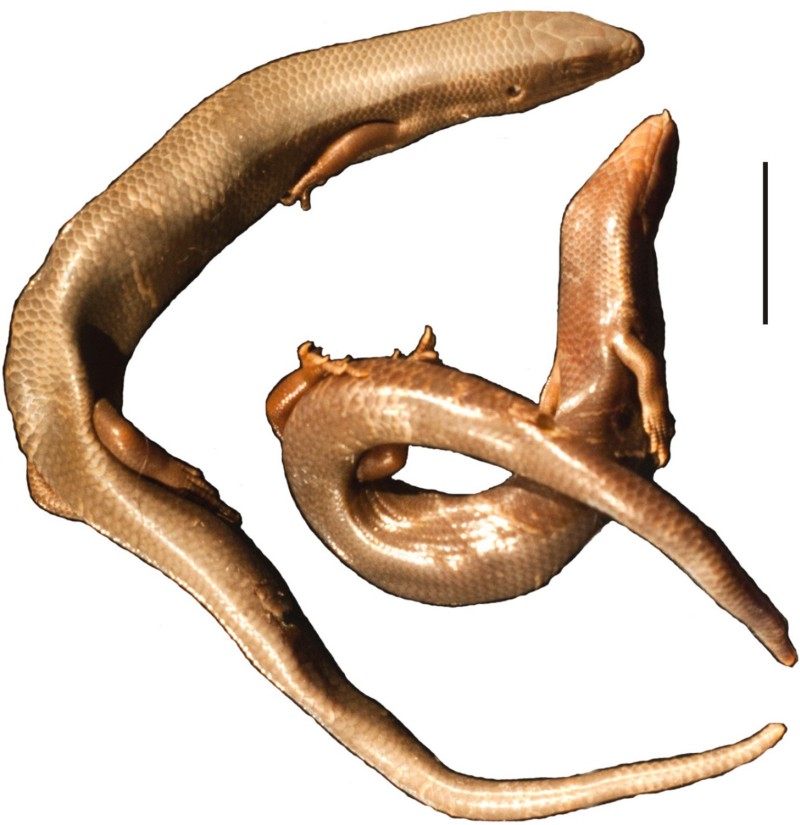

**Figure 14 Two specimens of *Chalcides viridanus* (*Gravenhorst, 1851*) (MNHUW uncatalogued) collected by Zimmer on Tenerife.** Scale bar = 1 cm. Photograph by Bartosz Borczyk.

uncatalogued; Fig. 14) and one of them could become the neotype. However, we refrain from making such a designation, because no molecular data could be obtained from any of these specimens. We recommend that a potential future neotype designation in *Chalcides viridanus* should involve specimen from a well-defined area (more precise than just 'Tenerife'), preferably also with molecular data, especially mitochondrial DNA, because there are several mitochondrial clades within this species.

## DISCUSSION

Natural history museums play a crucial role in the study of biodiversity. Redescriptions of historical specimens, especially name-bearing ones, housed in these institutions may have important implications for taxonomy and the nomenclature of many taxa (*Ohler & Dubois, 2016*), to some extent because many of the museum specimens are incorrectly identified and labelled (*Goodwin et al., 2015*). Such historical specimens served as the basis not only for recent redescriptions of important, name-bearing specimens (*Bucklitsch et al., 2012*; *Borczyk, 2013*; *Mecke et al., 2016*), but also for descriptions of species that are new to science, hitherto unrecognised (*Böhme et al., 2015*), and revalidations of species from synonymy (*Espinoza, Lobo & Etheridge, 2011*). The latter point may be especially important for diverse, species-rich groups

such as *Liolaemus*, with an often controversial taxonomy (*Lobo, Espinoza & Quinteros, 2010*; *Troncoso-Palacios et al., 2019*).

## Implications for *Liolaemus* taxonomy

We decided not to evert hemipenes in the rediscovered *Liolaemus* specimens because of their fragility which might lead to damaging them. The fact that precloacal pores were not observed in any of the specimens suggests that all of these individuals are females. However, members of the subgenus *Liolaemus* (a group to which all rediscovered lizards belong) tend to have significantly fewer precloacal pores than do members of *Eulaemus* (*Laurent, 1992*) and there are several *Liolaemus* species in which both females and males completely lack these pores (*Lobo, 2001*; *Pincheira-Donoso & Scolaro, 2007*; *Troncoso-Palacios et al., 2015b*). Also, the 'taphonomical' factor must be taken into consideration, that is, all the processes acting on a specimen after it was collected.

It seems unlikely that the species described by *Gravenhorst (1838)* represent valid, distinct species, given that they are known from single (*L. conspersus*, *L. hieroglyphicus*) or a few (*L. lineatus*) specimens and no new individuals have been reported for 180 years. However, there are several *Liolaemus* species that are currently considered valid and are known only from the type locality or only from the type specimen (*Meiri et al., 2018*), so this fact alone does not argue against their validity. They can be older synonyms of other species (see below) but this is difficult to test without molecular data, in part because of incongruence in the molecular and morphological phylogenies of the group (*Quinteros, 2013*; *Portelli & Quinteros, 2018*), and in part because interspecific hybridisation is common in *Liolaemus* lizards (*Olave et al., 2018*). Matters are further complicated by the fact that the type localities stated for Gravenhorst's taxa are usually very vague: 'Cauquenes' for *L. conspersus* and *L. hieroglyphicus* and 'Valparaíso' for *L. lineatus* (*Gravenhorst, 1838*).

To the best of our knowledge, none (except *L. lemniscatus*) of the *Liolaemus* nomina coined by *Gravenhorst (1838)* were used as valid names after 1899; however, this does not make them 'forgotten names', because only senior, and not junior, synonyms can be *nomina oblita* (*Ohler & Dubois, 2018*).

## CONCLUSIONS

We rediscovered several important specimens from the Gravenhorst's long thought to be lost herpetological collection at the University of Wrocław: type specimens of the teiid lizard *Callopistes maculatus* and liolaemids *Liolaemus conspersus*, *L. hieroglyphicus* and *L. lineatus*. Reexamination of the morphology of liolaemids revealed several taxonomically informative differences between these specimens and their presumed senior synonyms. While the synonymy of *L. lineatus* with *L. nitidus* is supported, the same cannot be done for *L. conspersus* and *L. hieroglyphicus* and their supposed senior synonyms, so we regard them as *species inquirendae*. Unfortunately, our attempts at molecular analyses were unsuccessful, so resolving the status of these taxa requires further, more complex studies. Nonetheless, the rediscovery of these important specimens underscores the importance of natural history collections, and their proper management

and protection, a point that was recently further strengthened by the tragic fire at the National Museum of Brazil in Rio de Janeiro in September 2018.

## ABBREVIATIONS

| | |
|---|---|
| **MNHNCL** | National Museum of Natural History, Santiago de Chile |
| **MNHUW** | Museum of Natural History, University of Wrocław |
| **SMF** | Senckenberg Forschungsinstitut und Naturmuseum Frankfurt |
| **SVL** | snout-vent length. |

## ACKNOWLEDGEMENTS

We thank Jan Kotusz and Tadeusz Stawarczyk (MNHUW) for access to specimens in their care, and Agnieszka Pietras-Lebioda for help with molecular analyses. We are grateful to Jaime Troncoso-Palacios, Luis Ceríaco and an anonymous reviewer for detailed and constructive comments that greatly improved the manuscript, and to John Measey for editing.

### Funding

The authors received no funding for this work.

### Competing Interests

The authors declare that they have no competing interests.

### Author Contributions

- Bartosz Borczyk conceived and designed the experiments, performed the experiments, analysed the data, contributed reagents/materials/analysis tools, prepared figures and/or tables, authored or reviewed drafts of the paper, approved the final draft.
- Tomasz Skawiński conceived and designed the experiments, performed the experiments, analysed the data, contributed reagents/materials/analysis tools, prepared figures and/or tables, authored or reviewed drafts of the paper, approved the final draft.

### Data Availability

All specimens described in this article are deposited in the Museum of Natural History, University of Wrocław (specimen numbers: MNHUW 1320, MNHUW 1321, MNHUW 1322, MNHUW 1323a, MNHUW 1323b and several unnumbered specimens).

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
