# Peer review of "Tracking down the lizards from Gravenhorst's collection at the University of Wrocław: type specimens of Callopistes maculatus Gravenhorst, 1838 and three Liolaemus species rediscovered"

_PeerJ, doi:10.7717/peerj.6525_

## Round 0.1 · original submission · Major Revisions

Three reviewers have read your submission, and all agree that it contains interesting and important information. It is appropriate for PeerJ. I agree with reviewers 2 & 3 that the manuscript could be better organised. Reviewer 2 gives you a useful template to work with, and I suggest that you follow this. All reviewers point out that the English is not of a standard that can be published, and I ask that you undertake a thorough revision prior to resubmission.

·

Basic reporting

no comment

Experimental design

no comment

Validity of the findings

no comment

Additional comments

This is a very interesting manuscript, I congratulate you and thank you for making this information available to all Liolaemus researchers. I recommend the following changes, especially in regard to Liolaemus lineatus.

Line 2: “of Callopistes maculatus”

Authors need to include the authority (name, year), change for: of Callopistes maculatus Gravenhorst, 1838,

Line 32-33: “Johann Ludwig Christian Gravenhorst (1777–1857) was the founder and the first director of the Zoological Museum at the University of Wrocław (currently Museum of Natural History).”

Please clarify that Gravenhorst was German, and also the original location of the Museum (Germany, later Poland?).

Line 57: “Pincheira-Donoso, Scolaro & Sura (2008)”

Change for Pincheira-Donoso et al. (2008)

Line 65: Check the citations in the entire document, “&” is used in lines 57-58 but “and” is used in the line 65. In some journals it is accepted use “&” for species authority, while “and” is used in the citations. Anyway, all citations need to use the same format through the manuscript.

Line 73: In the abstract it is stated that syntypes of Chalcides viridanus were not found but in line 73 is stated that DNA extraction was intended. Clarify in the line 73 that this last specimen of Chalcides viridanus is from other older collection.

Line 92: “Cauquenes Province in Chile”

I suggest change for: “Cauquenes in Chile (in the current Maule Region), collected by Herrn Scholtz”, because the word Province was not used by Gravenhorst.

Lines 92-93: “(currently Liolaemus fitzingeri)”

Change for (incorrect spell and use of the species authorities is needed): “(currently Liolaemus fitzingerii [Duméril & Bibron, 1837])”

Line 121: “Dorsal scales larger than dorsal neck scales but otherwise very similar (Fig. 3A)”

Check, Fig.3A, it actually shows the cloacal zone.

Lines 133-135: I seem to me that there are some dark scales between the shoulders, forming, maybe, what is left (discolored) of one inconspicuous occipital band, as is found in some Andean species from Maule Region. It is what I would expect after 150 years.

Line 141: Pincheira-Donoso, Scolaro & Sura, 2008).

Check “et al.” use in the entire document (only the species authorities should be mentioned one by one).

Line 146-148: This is a good argument. Whereas this character is very variable in some Liolaemus group, the nasal and the rostral scales are always separated in the L. nigromaculatus group, but also could be added that species of the L. nigromaculatus group are known to inhabit in the Antofagasta, Atacama, Coquimbo and Valparaíso regions (north to south), which is approximately 340 km (straight line) north from Cauquenes, you can cite TPet al.2015 (silvai).
Add that the fourth supralabial scale of L. conspersus is curved upward, and then, must be discarded that this species could be conspecific with Liolaemus fitzingerii, because this last belong to the Eulaemus subgenus, which has short uncurved supralabial scales (Lobo et al. 2010: A critical review).

Line 167: At this point of reading your very interesting manuscript, it is clear to me that conpersus synonymy with fitzingerii and nigromaculatus were two great misidentifications. I suggest you point out that L. conspersus, due to its SVL, 77 midbody scales, shape of the dorsal scales, and what maybe can be the “remains” of an inconspicuous occipital band, resembles some species of the Andean Maule Region, which is approximately 80 km to the east from Cauquenes. I refer to L. buergeri and L. flavipiceus, in which some specimen exhibit an inconspicuous occipital band. You can read about these in Garin et al. 2013 (Cuadernos de Herpetología 27 (1): 65-69), Pincheira-Donoso and Núñez 2005, Troncoso-Palacios et al. (2015, Liolaemus scorialis description). You can also state that L. conspersus does not match with the morphological features found among the species that occurs in Cauquenes or surroundings (chiliensis, cyanogaster, lemniscatus, schroederi, septentrionalis, tenuis) and cite PD&N2005 to argue about it.
However, I strongly recommend that you also point out that the holotype probably is not so well preserved to reach a clearly identification (buergeri or flavipiceus). At least to me, the best option is considered it a nomen dubium (uncertain application).
Maybe you should also add in the Line 97: “Undetermined sex”. Usually males have anal pores but these are absent in flavipiceus and are very small in buergeri, so 150 years later I think these could be discolored. Liolaemus can be easily sexed through gonads examination, but these specimens have survived WWII and too much else, it is not good idea open them for this...

Line 170-171: “from Cauquenes Province in Chile.”

Change for: “Cauquenes in Chile (in the current Maule Region), collected by Herrn Scholtz”

Line 172“(currently regarded as synonym of L. chiliensis; Pincheira- Donoso & Núñez, 2005)”

This need to be change for an older one: “(currently regarded as synonym of L. chiliensis; Donoso-Barros, 1966)”

Line 179: Expand and clarify, why it is different in unicolor?

Line 189: If possible, please add a close-up of the dorsal scales.

Line 206: Count the ventral scales.

Line 221: There are anal pores?

Line 235: I agree in that hieroglyphicus is not conspecific with L. lemniscatus. After this, add that the number of midbody scales allows to discard a synonymy with L. chiliensis, which has 31-33 (PD&N2005) and the fourth supralabial curved upward discard to signifer (Eulaemus subgenus, see previous comments).
L. hieroglyphicus is probably conspecific with schroederi or cyanogaster, also it resembles septentrionalis juvenile. This need to be added, see features in PD&N2005 and record of cyanogaster very near Cauquenes in Rubio et al. 2004 Herpetological Review 35(3). However, L. septentrionalis has rounded dorsal scales with no mucron. The Fig. 4 resolution does not allow me check it. These three species should be mentioned as hypothesis but more accurate studies are needed.

Line 251: Clarify that Fitzinger (1843) does not refer to L. lineatus main variety (Hauptart), while he synonymize the three Abarten.

Line 254: “Boulenger (1885) synonymised this species with L. nitidus”

Clarify: Later, Boulenger (1885) synonymized these four varieties of L. lineatus with L. nitidus.
Note that he explicit indicates Fig. 1-7 by Grav. as L. nitidus.

Line 280: If possibly, count the scales around midbody. Only two Liolaemus with this type of dorsal scales occurs in Chile, L. chiliensis and L. nitidus. Both species can be easily differentiated because the throat of nitidus has dark stripes whereas chiliensis always has whitish, without stripes, throat. For me, this is by sure a L. nitidus specimen.

Line 289: I agree, this is a problematic specimen. Stripes in the throat and the dorsal scale shape-size, indicate L. nitidus juvenile. But frontal fragmentation, dorsal scale shape-size indicate L. chiliensis. Both species has nasal and rostral scales in contact with apparent no intraspecific variation (my own data, you can cite me as personal comment).

Line 314-315: “However, its phylogenetic position in molecular analyses is highly variable (Troncoso-Palacios et al., 2015; Panzera et al., 2017; Portelli & Quinteros, 2018),”

This is incorrect. While Troncoso-Palacios et al. 2015 and Panzera et al. 2017 are based in molecular data, Portelli & Quinteros 2018 performed a combined phylogeny with DNA + morphology data. This need to be clearly stated.
Some researchers will conclude that Portelli & Quinteros (2018) result should be taken with caution because data combination can amplify the errors (https://academic.oup.com/sysbio/article/63/4/582/1623176).

Line 321-323: “The matters are further complicated by the fact that L. nitidus is probably a paraphyletic species, with some of its lineages being more closely related to some lineages of L. monticola (which probably is also paraphyletic; Torres-Pérez et al., 2017).”

I disagree. This sentence suggests that L. nitidus is composed by more than one species. I think that this is not what is really concluded in Torres-Pérez et al. (2017). Indeed they point out that nigroviridis and monticola appear to be composed by several hidden species, but in the case of nitidus, although they found that one haplotype of nitidus is nested in monticola northern race and two in monticola southern race, they found eight haplotypes forming the main nitidus clade (only one species), and this main clade even includes samples from the same localities where outsider haplotypes were detected (Cantillana mountain). They suggest incomplete lineage sorting and introgression as explanation for the paraphyletic result but they did not suggest that nitidus is composed by more than one species.
I recommend delete this sentence.

Line 330: “such as presence of two frontal scales rather than one”

It is very interesting, however, although I have no reviewed nitidus specimen with two frontal, I will attach some photograph with variation on this.

Line 331: “and contact between nasal and rostral scales.”

This is interesting. This character is informative for species of the L. nigromaculatus, but it is variable among other groups and even in the same species. However, all my samples of nitidus (n=16), all have these in contact, which suggest that this feature is fixed also in nitidus.

Line 331-333: “It is possible that this specimen represents different lineage of L. nitidus but without molecular data it is currently impossible to evaluate.”

It is too much speculative. As mention before, only some sequences from Cantillana Mountain are nested in monticola (northern and southern races), while other samples from Cantillana and all other nitidus samples are nested in nitidus main clade. This suggests introgression or incomplete lineage sorting but not several “nitidus” species. At least it requires more sampling, but I don’t agree with speculate this based in such problematic specimen. I recommend delete this sentence.

Line 341: After this, explain that L. lineatus Gravenhorst 1838 must not be confused with L. lineatus Gray, 1845, an invalid name (previously occupied) of the species currently knows as L. nigroviridis (see Núñez 2004, Cambios taxonómicos para la herpetofauna de argentina, bolivia y chile).

Line 380: “represents L. nitidus or a very similar species.”
Again this suggests several species in nitidus, which has not been found in phylogenetic studies (paraphyletic samples came from only one locality). I suggest delete or rewrite as “L. nitidus or L. chiliensis”

Line 392: Add the type locality: “at foot of Cordillera, Chile”. By the way, this is a very problematic type locality. There is a label with more information?

Line 397: After this, should be included that previously (ignoring the survival of Gravenhorst specimen), Veloso et al. (2000, Fixation neotype lacerta palluma) intend the designation of the MNHNCL 2909 as the neotype of Lacerta palluma (currently C. maculatus), but this proposal was rejected by Etheridge and Savage (2003) and by the International Commission of Zoological Nomenclature (ICZN 2005, case 3225, International Commission on Zoological Nomenclature. 2005. Opinion 2118 (Case 3225). Phymaturus Gravenhorst, 1837 and Lacerta palluma Molina, 1782 (currently Phymaturus palluma, Reptilia, Sauria): usage of the names conserved by the designation of a neotype for Lacerta palluma Molina 1782. Bulletin of Zoological Nomenclature 62:116–117.).

·

Basic reporting

Both authors are not native speakers (neither I am), and this denotes through the text. There are lot of problems with the English, namelly in phrase structure and grammar. I suggest that the authors ask a native speaker to review the English before the next subimission.

The organization of the paper is problematic. The authors need to follow an example from other existing type catalogs. None of the nomina have their taxonomic authorities or date associated, the type localities are not in verbatim, etc.

The other main problem of the manuscript is that, without much explanation, the authors present some brief taxonomic accounts for other taxa, namelly Liolaemus lemniscatus, Liolaemus marmoratus, Liolaemus unicolor, Leiocephalus schreibersii, and Chalcides viridianus. All of these taxa were named by Gravenhorst, but besides that, there is nothing about them that deserves to be noted in the paper, besides the fact that the types are thought to be lost and the authors couldn't find them.

The authors have to decide what they want to present in this manuscript - it is either a paper about the specimens they rediscovered, and then stick to those only, or if they want to present a paper about taxa named by Gravenhorst, and then expand it to include taxonomic accounts of all of them, with or without types.

There is also a lot of inconsistencies that complicate the manuscript - for example, in the case of Liolaemus lineatus, the authors refer that Gravenhorst described one main variety and three subvarieties. In the following sentences the authors call the subvarieties varieties, and start to refer to the specimens by "first specimen" and "second specimen".

This is highly confusing. I understand that both specimens have the same catalog number, but in cases like these each one should be identified separately in a non-confusing way - maybe call one 1323a and the other 1323b, adding a tag to the respective specimen.

I suggest the authors reformat the paper, for example, following my 2014 type catalog of Porto Museum
(Ceríaco, L.M.P. et al. (2014) Catalogue of the amphibian and reptile type specimens of the Museu de História Natural da Universidade do Porto in Portugal, with some comments on problematic taxa. Alytes 31(1/2): 13 - 36)

This is what it would look like:

Extant types

Reptilia

Family

Genus species Author (Date: page of description)

Type specimens: XX holotype/syntypes/paratypes designated by Gravenhorst (date of publication) still exist in the Museum name, being represented by one juvenile/adult male/adult female/unsexed adult specimen from “Verbatim locality” [=current name of the place], other geographic data (province, region, etc), Country, collected in DATE of Collection (if known) (Catalog number), [in the case of more specimens from different places] and add same data as above and then (Catalog number).

Present name: Genus species Author (Date) [according to the most updated or followed taxonomy]

Remarks: in here add everything you have about the species and specimen (now divided by a smaller unnamed introduction, "Redescription of the holotype" and "Discussion"). Maybe start with the taxonomic and nomenclatural story, and then follow with the type material redescription. Do not separate it into different subsections.

Experimental design

There is nothing to add regarding the experimental design itself. It is a straightforward paper dealing with lost types. The conclusions and identification of the specimens are well supported.

Validity of the findings

The discovery and redescription of the types that everyone thought to be lost is an important and valid finding. The data and proofs that the author's show support the unambiguous identification of the types.
However, as noted in my comments in the pdf, there are some problems with the lectotype designations, namelly the designation of a lectotype for Callopistes maculatus, or the designation of a lectotype for L. lineatus main variety using the putative type of one of the subvarieties.

Additional comments

This is a very interesting and important paper, but it has a lot of formatting and language problems. I encourage you to revise the format, and especially make it more consistent. I also suggest that you stick with the specimens you have discovered, and avoid talking much about the ones you didn't. The Chalcides part should be deleted, is highly irrelevant.
As my identity is revealed to the authors, I'm open to any querie and more direct help if they wish to contact me.

Reviewer 3 ·

Basic reporting

The manuscript is well written in general but I found some typo and format mistakes that must be corrected (see below). The background included in the text allows understanding how this work fits in the broader field of knowledge. Objectives aren’t included; someone can guess what the aim of the study is. The relevant prior literature is appropriate referenced. Nevertheless, some relevant literature is missing. The general structure of the article is odd. Authors joined results and discussion into a single section. This is not a good choice, since some nomenclatorial acts are proposed. So results and discussion should be located as separate sections. The study is interesting, but in the way that the study was framed, I can’t see the relevance. Authors only take character states of specimens; comparisons are based on literature (since no specimens list are detailed), and no formal re-descriptions are made.

Experimental design

My main concern is about the lack of specimens studied. I understand the difficulty of study some material, but in cases where nomenclatorial and taxonomic results arise, it’s necessary to study the characters from specimens (not based on literature). Especially when in figures I count a different number of scales that the authors made (see below). Perhaps some character states were taken in different way. Authors mentioned that they “focused on describing morphological characters used by previous authors...” There are a lot of character states that authors don’t include. Those can be very useful for comparisons. Author miss scales counts: Number of temporal scales, number of gular scales, number of neck’s scales, Hellmich’s index, among others.

Validity of the findings

Re-descriptions are good. Nevertheless, there are many character states which were taken for one specimen and not for the others. Also, for each specimen redescribed, SVL and sex are lacking. SVL is on Table, but must be in redescription. Authors mention large and robust lizard without mention of SVL. About sex, authors mentioned at the end of the MS that the specimens’ could be females (because de lack of precloacal and femoral pores). Data about sex must be included at begin of the descriptions.
Comparisons were made against single species (except for L. conspersus), and based on so few characters. If the aim of the study is to re-describe the species, a complete diagnosis is necessary; especially for lectotype of L. lineatus.
Why authors report number of infradigital lamellae of third finger? The character state traditionally used is fourth finger.
I count different numbers of supralabials, lorilabials and infralabials (see below).
As I mentioned above, comparisons were mostly made with single species, mainly following Boulenger synonymies. Lines 92-93 authors mentioned that L. conspersus was synonymized with the current L. fitzingerii. Despite this, authors only compare and discuss the taxonomy position of L. conspersus against L. nigromaculatus and the species related. Why not compare it with L. fitzingerii? Just, should look by the presence (or absence) of femoral patch. Similar occurs with L. hieroglyphicus. Authors compare it with L. lemniscatus. But previously they mentioned that this taxon was synonymized with L. signifer (line 171) and L. chiliensis (line 172). Picture of L. hieroglyphicus is far from being a L. signifer, but is morphologically very close to L. chiliensis. So, comparisons with this latter taxon are needed. Since authors compare L. hieroglyphicus with L. lemniscatus based on literature, must include data taken from Quinteros (2012).
Designation of lectotype and paralectotype must follow The Code. There is not citation of the ICZN on the MS. Moreover, if authors propose a lectotype (and paralectotype) a formal description must be included. Many section of a formal description lack.
The “mysterious specimen” is anecdotic. If the specimen is uncatalogued, how authors know that was a Gravenhorst’s specimen? Only Gravenhorst collect Liolaemus specimens for the Museum?.
As for Liolaemus lineatus, I expect a formal description for Callopistes maculatus.
About General comments and implications in Liolaemus taxonomy. Even though there are many species of Liolaemus (and many other genera) based only on specimens from the type locality or the type series, the main concern about Gravenhorst’s species is the number of individuals and the early synonym proposed.
Te comments about asymmetry and mosaic characters’ distribution as evidence of hybrids has no relation with the MS.
If authors want to publish this study, should include data for many more species of Liolaemus, in order to stablish the taxonomic status of the specimens of Gravenhorst. Should perform a formal redescription of Callopistes maculatus, including additional specimens.

Additional comments

I have some minor comments (mainly about format):
Line 55.- Spell out Liolaemus. If the genus name begin a sentence or if is named for first time in a new paragraph, must be spelled completely. There many of this mistakes along the MS.
Line 97.- Include SVL and sex. Please make this changes for all specimens redescribed.
Lines 107-108.- I follow the figures and count different number of supralabials: I count five supralabials (from the right side). And the fourth is upturned but not contacting the subocular scale.
Lines 108-109.- I count six loriabials, three in contact with the subocular.
Lines 114-116.- Include number of temporal scales
Lines 119-120. Sentence about neck scales. Please re-write
Lines 121 and 130.- Figure 3B indicate dorsal scales, and Figure 3A show the lack of pores.
Line 125.- Why third finger? Ussually the infradigital llamellaes are counted from the fourth finger/toe.
Line 130.- No Liolaemid lizard exhibit femoral pores
Most of the previous comments are repeated for all taxa. Especially the scales count (lorilabials, supra and infralabials).
Line 201.- Probably is the figure, but upper temporal show a slight keel?
Line 204.- Provide number of gular scales (for all taxon), when possible.
Line 206.- What about number of dorsals and ventrals?
Lines 246-247.- Sentence is hard to follow. Please re write
Lines 273-274.- I count the same number of lorilabials. I don’t know how authors made the previous count.
Line 306.- “anteriormost nuchal scales” are you talking about occiput region??
Line 317.- “computational method”. For systematic these are Optimality Criterion.
Line 405.- Authors must mention sex of the specimen, at the begin of the description.

---

## Round 0.2 · Minor Revisions

I have received two further reviews of your revised manuscript which suggest that contingent on some minor revisions, your manuscript should be acceptable for publication.

Please consider the suggested changes, especially with respect to removing the 'Unidentified Specimen' section.

Please also be aware of a recent publication on liolaemid taxonomy:

Without a body of evidence and peer review, taxonomic changes in Liolaemidae and Tropiduridae (Squamata) must be rejected: Zookeys
https://zookeys.pensoft.net/articles.php?id=29164

·

Basic reporting

no comment

Experimental design

no comment

Validity of the findings

no comment

Additional comments

This time it suggests few corrections, mainly drafting issues. It is a very interesting manuscript, I hope it will be published son.

Line 54-57: “Here we redescribe other type specimens of lizards that were rediscovered in the collections of the Museum of Natural History, University of Wrocław. We also discuss taxonomic implications of these findings and provide a catalogue of type specimens of all lizard species named by Gravenhorst.”
Move after line 65, as final paragraph of the Introduction. Otherwise, the lines 58-65 seems disconnected.

Line 109: add: collected by Scholtz.

Line 118: “the name”
Change for: these names (Callopistes maculatus and Phymaturus palluma) became universal.

Lines 119-120: “attempted to synonymise C. maculatus with P. palluma and proposed a neotype (MNHNCL 2909)”
Correct, I suggest: Veloso, Núñez & Cei (2000) proposed the combinations C. palluma and P. flagellifer (formerly Centrura flagellifer), and attempted to synonymise C. maculatus with C. palluma, designating a neotype (MNHNCL 2909) for the last.

253-254: “Interstitial granules are present between the dorsal scales, as in some L.
nigromaculatus and L. atacamensis, but not in L. silvai and L. zapallarensis.”
Clarify, in regards to L. conspersus.

255-256: “Liolaemus conspersus has significantly more scales around the midbody than any of the above-named lizards (77), while in the others the range is 48–62 scales (Troncoso-Palacios & Garin, 2013”
I suggest: Liolaemus conspersus has significantly more scales around the midbody (77) than any of the above-named lizards (48–62 scales, Troncoso-Palacios & Garin, 2013)

264-269: “It is also worth noting that the phylogeny and taxonomic content of the nigromaculatus group are not well established. Mitochondrial DNA analysis by Troncoso-Palacios et al. (2015a) indicates that all these species are closely related, but Panzera et al. (2017), on the basis of analyses using 541 ultra-conserved elements and 44 protein-coding genes, suggested that L. atacamensis is only distantly related to the other named species, even though it is morphologically most similar to L. nigromaculatus (Troncoso-Palacios & Garin, 2013).”
I suggest to delete this paragraph, because it is not a contribution to clarify the taxonomy of L. conspersus.

278: “the Maule region”
Change for: Maule Lagoon

412: You can also add that L. gravenhorstii is restricted to the Metropolitan region of Chile, more than 200 km north from Cauquenes.

423: I suggest you restrict this proposal to L. schroederi and L. cyanogaster, because L. gravenhorstii midbody scale count and distribution strongly suggest that it is not a synonym of L. hieroglyphicus.

425: Check this for L. conspersus and L. hieroglyphicus. If you think that —due to the state of the holotype—it is unlikely that a further study can be succeed in establishing the taxonomic identity, then I suggest nomen dubium, but if the status of preservation of the holotype is appropriate, then I suggest nomen inquirenda.

447: “Unfortunately, for unknown reasons, the first variety was not illustrated”
I suggest: Unfortunately, the first variety was not illustrated

448-450: “However, the survey of two museum catalogues that survived the war supports that, at least originally, four specimens of L. lineatus were present (later updated to two
specimens), In the second catalogue (from 1907), three specimens are listed (Fig. 9).”
This was hard to read for me. Later updated to two specimens? (after 1907?, move at the end of this sentence if that is the case). Check the “, In”, should be “. In”.
Maybe something like this: However, the museum catalogue before 1907 supports that, at least originally, four specimens of L. lineatus were present, while a second catalogue (from 1907) list three specimens (Fig. 9).

462: “Liolaemus lineatus Gravenhorst, 1838 should not be confused with its younger homonym Liolaemus lineatus Gray, 1845, a species currently regarded as synonymous with Liolaemus nigroviridis Müller & Hellmich, 1932 (Núñez, 2004).”
Move to the line 457, after “…with L. nitidus.” I suggest: Boulenger (1885) synonymised this species (all four varieties) with L. nitidus. We remark that Liolaemus lineatus Gravenhorst, 1838 should not be confused with its younger homonym Liolaemus lineatus Gray, 1845, a species currently regarded as synonymous with Liolaemus nigroviridis Müller & Hellmich, 1932 (Núñez, 2004).

687: “Liolaemus (also called the chiliensis section – a group to which all rediscovered lizards belong)”
Correct, I suggest: Liolaemus – a group to which all rediscovered lizards belong–
This because each Liolaemus subgenus is composed of two sections. Liolaemus sensu stricto is composed of chiliensis and nigromaculatus sections, while Eulaemus is composed of linieomaculatus and montanus sections. By sure all the types rediscovered belongs to the Liolaemus sensu stricto, but it is not necessary to point out the section.

·

Basic reporting

The manuscript is much more improved and almost ready for publication. I would still reconsider the "Unidentified Specimen" section - at least do not make it a separate subsection but simply continue the text under the remarks subsection, immediately after the last paragraph.
Not sure if in the case of Liolaemus conspersus and L. hieroglyphicus the authors should say that the present name is respectively Liolaemus conspersus (Line 179) and L. hieroglyphicus (Line 308). The authors present a very nice overview of the taxonomic and nomenclatural problems, and realize that maybe none of the current synonymies is correct. However, I would be conservative and follow the most recent revision of the group and maybe just call it "cf." nigromaculatus and cf. lemniscatus. The remarks then explain the problems.

Experimental design

All ok for such kind of study.

Validity of the findings

While I'm not an expert on these taxa, the authors present good comparisons, and the comments of the other reviewers, Jaime Troncoso-Palacios seems to confirm that.

Additional comments

The manuscript is much more improved and almost ready for publication. I would still reconsider the "Unidentified Specimen" section - at least do not make it a separate subsection but simply continue the text under the remarks subsection, immediately after the last paragraph.
Not sure if in the case of Liolaemus conspersus and L. hieroglyphicus the authors should say that the present name is respectively Liolaemus conspersus (Line 179) and L. hieroglyphicus (Line 308). The authors present a very nice overview of the taxonomic and nomenclatural problems, and realize that maybe none of the current synonymies is correct. However, I would be conservative and follow the most recent revision of the group and maybe just call it "cf." nigromaculatus and cf. lemniscatus. The remarks then explain the problems.

---

## Round 0.3 · accepted · Accept

Thanks for your new version of the ms.

Please check your proofs carefully. There might be some odd grammar that needs attention. E.g. L76: "His main interests lied in.." should be "His main interests lay in..."

#